# DeepDOF-SE: affordable deep-learning microscopy platform for slide-free histology

Lingbo Jin[1,6], Yubo Tang[2,6], Jackson B. Coole[2], Melody T. Tan [2], Xuan Zhao[1], Hawraa Badaoui[3], Jacob T. Robinson [1], Michelle D. Williams [4], Nadarajah Vigneswaran[5], Ann M. Gillenwater [3], Rebecca R. Richards-Kortum [2] ✉ & Ashok Veeraraghavan [1] ✉

Histopathology plays a critical role in the diagnosis and surgical management of cancer. However, access to histopathology services, especially frozen section pathology during surgery, is limited in resource-constrained settings because preparing slides from resected tissue is time-consuming, labor-intensive, and requires expensive infrastructure. Here, we report a deep-learning-enabled microscope, named DeepDOF-SE, to rapidly scan intact tissue at cellular resolution without the need for physical sectioning. Three key features jointly make DeepDOF-SE practical. First, tissue specimens are stained directly with inexpensive vital fluorescent dyes and optically sectioned with ultra-violet excitation that localizes fluorescent emission to a thin surface layer. Second, a deep-learning algorithm extends the depth-of-field, allowing rapid acquisition of in-focus images from large areas of tissue even when the tissue surface is highly irregular. Finally, a semi-supervised generative adversarial network virtually stains DeepDOF-SE fluorescence images with hematoxylin-and-eosin appearance, facilitating image interpretation by pathologists without significant additional training. We developed the DeepDOF-SE platform using a data-driven approach and validated its performance by imaging surgical resections of suspected oral tumors. Our results show that DeepDOF-SE provides histological information of diagnostic importance, offering a rapid and affordable slide-free histology platform for intraoperative tumor margin assessment and in low-resource settings.

Over 19 million new cancer cases were diagnosed worldwide in 2020[1]. Surgery is one of the most common treatments for many types of cancer[2], and the goal of surgery is to remove all cancer, while preserving normal tissue to minimize loss of function. This requires intraoperative tumor margin assessment to delineate between healthy and cancerous tissue, where resected specimens are immediately frozen, and sectioned in a cryostat microtome, and stained for microscopic examination. In addition to frozen section pathology during surgery, cancer is routinely diagnosed with histopathological examination that requires the specimen to be fixed in formalin, dehydrated in increasing concentrations of alcohol, embedded in paraffin, and then thinly sectioned with a microtome. In both formalin-fixed paraffin-embedded

[1]Department of Electrical and Computer Engineering, Rice University, 6100 Main St, Houston, TX, USA. [2]Department of Bioengineering, Rice University, 6100 Main St, Houston, TX, USA. [3]Department of Head and Neck Surgery, University of Texas MD Anderson Cancer Center, 1515 Holcombe Blvd, Houston, TX, USA. [4]Department of Pathology, University of Texas MD Anderson Cancer Center, 1515 Holcombe Blvd, Houston, TX, USA. [5]Department of Diagnostic and Biomedical Sciences, University of Texas Health Science Center at Houston School of Dentistry, 7500 Cambridge St, Houston, TX, USA. [6]These authors contributed equally: Lingbo Jin, Yubo Tang. ✉e-mail: rkortum@rice.edu; vashok@rice.edu

(FFPE) and frozen section, microscopic examination of subcellular features necessitates a slide preparation process that removes surface irregularities, minimizes subsurface scattering, and improves imaging contrast. However, current slide-based histology workflows are time- and labor-intensive and require expensive laboratory infrastructure along with highly trained histotechnologists, making them inaccessible in resource-constrained settings[3,4]. To bypass the complex histopathology preparation process, a slide-free approach is desired to quickly provide histologic quality images of fresh tissue specimens at the point of resection without expensive equipment.

Any slide-free approach to real-time histopathology at the point of resection must somehow (a) reduce sub-surface scattering without the need for thin sections, (b) contend with natural surface irregularities without requiring use of a microtome and (c) retain the perceptual appearance of traditional processing to facilitate its integration in the routine clinical practice. We developed the deep-learning-enabled extended depth-of-field (DOF) microscope with UV surface excitation (DeepDOF-SE) to achieve these goals, by substantially expanding the imaging capability of a simple dual-channel fluorescence microscope with integrated computational and deep learning models. DeepDOF-SE is specifically designed to provide a slide-free histology platform for use in low-resource settings to support immediate biopsy assessment and/or rapid intraoperative assessment of margin status. The 4×, 0.13 NA system can resolve subcellular features needed to diagnose pre-cancer and cancer and is consistent with pathologists' use of 2× and 4× objectives for the vast majority of diagnoses[5–8] and recent studies demonstrating that deep learning models can accurately classify the presence of cancer with significant image compression[9] and NA as low as 0.05[10].

Over the last decade a host of novel techniques have been developed for optical sectioning of thick tissue samples to suppress sub-surface scattering. Two-photon microscopy[11], confocal microscopy[12], light-sheet microscopy[13], optical coherence microscopy[14], and photoacoustic 3D microscopy[15] have been shown to achieve optical sectioning and enable high-contrast imaging of thick tissues. However, all these techniques require complicated and expensive opto-mechanical components that are difficult to align and calibrate, and as such fail to naturally lend themselves to use in point-of-resection and/or low-resource settings[16]. For instance, the light source alone used in an open-access light-sheet microscope[17] costs more than the entire DeepDOF-SE system (less than $7000). In DeepDOF-SE, we aim to provide histology of tissue surfaces cut with a simple scalpel, and we rely on microscopy with UV surface excitation (MUSE)[18] which exploits the limited depth of penetration of UV excitation light to limit fluorescent emission only to the tissue surface—thereby limiting the deleterious effects of sub-surface scattering. Thus, UV excitation allows us to reduce sub-surface scattering without the need for thin sections.

Computational imaging and especially end-to-end optimized optics, sensors and algorithms have emerged as a powerful tool for achieving performance beyond that of conventional optics. For example, single-shot 3D microscopy[19–22], extended depth-of-field microscope[23], and lensless microscopes[24–27] have shown great potential for high-resolution imaging using simple and compact systems. The key ingredient in all these techniques is that co-designing optics and algorithms allows these systems to overcome the limits of conventional optics. However, few computational imaging techniques have so far been designed to image dense cellular features as observed in histology. In DeepDOF-SE, we build upon our previously developed deep learning framework to extend the microscope depth-of-field by co-designing wavefront encoding and image processing[23]. Compared to our previous work on extended depth-of-field imaging at a single wavelength, the end-to-end optics and image processing design in DeepDOF-SE is optimized in two fluorescence channels. Here we demonstrate the capability of DeepDOF-SE to image nuclear and cytoplasmic features simultaneously, and its compatibility with

different fluorescence dyes across a broad range of emission wavelengths. Moreover, we show that information acquired in two fluorescence channels allows seamless integration of deep-learning-based virtual staining to generate H&E-like histology images.

Generative artificial intelligence (AI) has made remarkable advances over the last few years, resulting in a wide variety of powerful algorithms for virtual staining of histology images. Unlike conventional analytical virtual staining methods[28], generative adversarial network (GAN) based virtual histology does not require user input or prior knowledge of the stains' property. Supervised GANs built on Pix2pix[29] can virtually stain label-free slides or translate the appearance of a slide chemically stained with one dye to mimic that of another stain or combinations[30,31]. While these supervised models have few artifacts, training them requires a large amount of paired data, which is expensive to acquire. Unsupervised CycleGAN-based[32] virtual staining can be trained on unpaired images[33], making it one of the most frequently used deep learning frameworks for different histological applications, including label-free virtual staining, stain-to-stain transformation, and correction of stain variations[31,34]. In virtual staining applications, CycleGAN was evaluated with different imaging modalities, such as MUSE and photoacoustic microscopy, and virtual staining has been demonstrated in different tissue types including brain, breast, prostate, and bone specimens[15,33,35]. In DeepDOF-SE, we demonstrate a two-step semi-supervised scheme to train the Cycle-GAN for virtual staining, generating artifact-free virtual H&E while avoiding the need for acquiring paired data. This framework is readily applicable to different staining protocols that provide both nuclear and cytoplasmic contrast. Furthermore, we report the application of CycleGAN for virtual staining of fresh human oral tumor resections, and we demonstrate that our model is capable of visualizing distinct histological features in different layers of oral epithelium.

In this work, we report the development of DeepDOF-SE by combining surface excitation, extended DOF imaging, and virtual staining; we build the DeepDOF-SE based on a simple dual-channel fluorescence microscope and demonstrate its use for rapid histological imaging of fresh intact tissue specimens at a scanning speed of 1.6 cm²/min. Building upon our previous work on extended depth-of-field microscopy relying on a single dye to provide contrast, the current work now incorporates optical sectioning[18], deep-learning-enabled extended DOF imaging and virtual staining of nuclear and cytoplasmic features to make rapid, cost-effective, and slide-free histology practical. As shown in Fig. 1a, DeepDOF-SE is based on a simple dual-channel fluorescence microscope. Image contrast is provided by briefly immersing fresh tissue samples in a solution containing the vital fluorescent dyes Rhodamine B and DAPI that highlight cytoplasmic and nuclear features, respectively. A jointly optimized phase mask and reconstruction network extend the depth-of-field, enabling high-resolution images to be collected without refocusing from tissue surfaces that are simply cut with a scalpel; compared to our previous work on a single-channel extended DOF microscope[23], DeepDOF-SE provides the ability to simultaneously acquire images of cytoplasmic and nuclear features in two separate channels. The resulting all-in-focus fluorescence images are virtually stained to resemble H&E-stained sections using a semi-supervised CycleGAN. Figure 1b demonstrates the improvement in performance provided by surface excitation, extended DOF, and virtual staining. Compared to visible excitation (dual-channel fluorescence image of porcine specimen stained with Rhodamine B and DAPI in Fig. 1b.i), UV excitation suppresses the impact of subsurface scattering[18] (Fig. 1b.ii); however, it is challenging to survey a large area (1 cm² or larger) of scalpel-cut tissue due to surface irregularities that extend beyond the DOF of a conventional microscope, as evidenced by out-of-focus regions of the image. We tackled this challenge by using deep-learning techniques to extend the DOF to 200 μm, consistent with topographic variations in tissue prepared with a simple scalpel[13] (Supplementary Fig. 2), while preserving

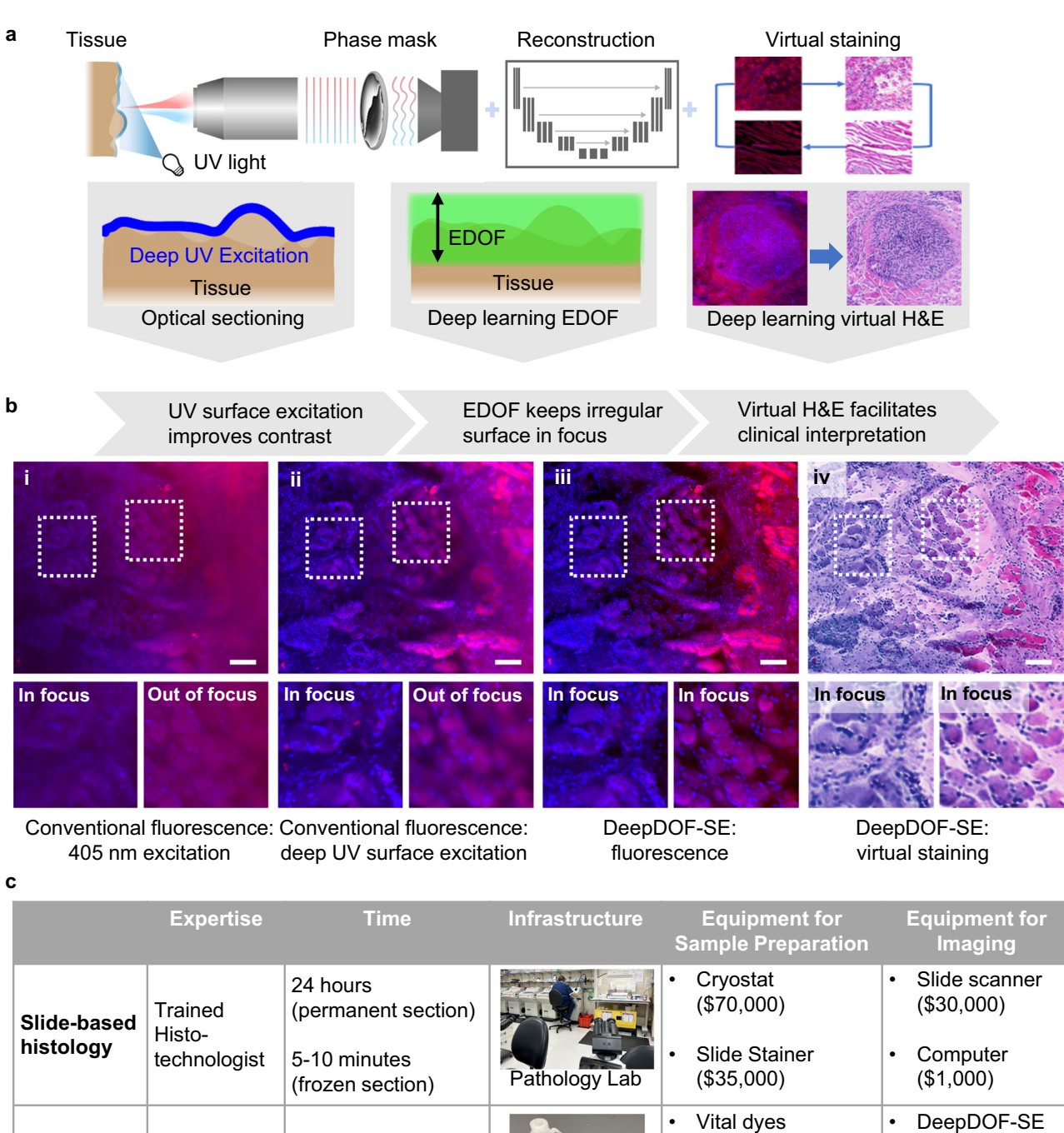

**Fig. 1 | Overview of the DeepDOF-SE platform for slide-free histology of fresh tissue specimens. a** DeepDOF-SE is built based on a simple fluorescence microscope with three major components: surface UV excitation that provides optical sectioning of vital-dye stained fresh tissue; a deep-learning-based phase mask and reconstruction network that extends the depth-of-field, enabling in-focus imaging of irregular tissue surfaces; and a CycleGAN that virtually stains fluorescence images resembling H&E-stained sections. **b** Compared to a conventional fluorescence microscope, DeepDOF-SE acquires high-contrast, in-focus and virtually stained histology images of fresh tissue specimens. Example images of an ex vivo porcine tongue sample were acquired using (i) a conventional fluorescence microscope with 405 nm excitation, (ii) a conventional fluorescence microscope with 280 nm excitation, (iii) DeepDOF-SE in fluorescence mode, and (iv) DeepDOF-SE with virtual staining. Benefits of optical sectioning, extended depth-of-field, and virtual staining are shown from left to right with the addition of UV excitation, deep-learning-enabled extended DOF imaging, and CycleGAN-based virtual staining. Scale bars are 100 μm. Brightness increased for display. **c** Compared to conventional histopathology, DeepDOF-SE significantly reduces the time, infrastructure, and expertise needed to prepare histology samples. UV ultraviolet, DOF depth-of-field, H&E hematoxylin and eosin, DeepDOF-SE a deep learning enabled extended depth-of-field microscope with surface excitation.

subcellular resolution[23]. As shown in Fig. 1b.iii, the combination of UV excitation and deep-learning extended DOF allows acquisition of an in-focus image from a large area. Finally, Fig. 1b.iv displays the CycleGAN virtual H&E stain of the image, designed to resemble conventional slide-based H&E. Using a data-driven and learning-based approach, we show here that DeepDOF-SE offers a slide-free histology platform suited for rapid histopathologic assessment of fresh tissue specimens that could be performed intraoperatively or in resource-constrained settings (Fig. 1c).

## Results

### End-to-end design for extended DOF fluorescence imaging

The DeepDOF-SE fluorescence microscope enables direct in-focus imaging of irregular surfaces of fresh tissue specimens by extending the depth-of-field and leveraging surface excitation. Figure 2 describes the end-to-end network used to jointly design the phase mask and the reconstruction algorithm. The first layer of the end-to-end network uses a physics-informed algorithm to simulate image formation of a fluorescence microscope with the addition of a phase mask. In particular, image formation at two spectral channels that correspond to the vital dyes, Rhodamine B and DAPI, is simulated at 21 discrete depths within the 200 μm DOF. In the following layers of the end-to-end network, two reconstruction U-Nets are used to recover all-in-focus images from the blurred images.

After the network was trained with a dataset containing a broad range of complex features including histologic features (see "Methods"), the optimized phase mask design was fabricated and installed in the DeepDOF-SE microscope. Figure 3a shows the system design based on a simple fluorescence microscope with a standard objective (Olympus Plan Fluorite 4×, 0.13 NA). A UVC LED provides oblique illumination for surface excitation, while the phase mask modulates the wavefront in the optical path to enable the extended depth-of-field imaging. We performed a one-time calibration of the system by capturing its point spread functions (PSFs, shown in Fig. 3b) and used the measured PSFs to fine-tune the U-Nets.

### Validation of DeepDOF-SE performance in the target DOF

We characterized the spatial resolution of DeepDOF-SE using a negative 1951 USAF resolution target. In Fig. 3c, the resolution target was imaged in two fluorescence channels (Rhodamine B and DAPI) using DeepDOF-SE and a conventional fluorescence microscope. As shown in Fig. 3c, significant defocus blur was observed as the USAF target was translated axially through the focal plane of the conventional microscope. In contrast, Group 7 element 5 (2.46 μm line width) is consistently resolved in the Rhodamine B and DAPI fluorescence channels of DeepDOF-SE as the target is translated axially through the target 200 μm depth-of-field. Notably, we also observed significant axial chromatic aberrations between the two fluorescence channels using the conventional microscope, which can further hinder direct imaging of uneven surfaces; using the DeepDOF-SE, the chromatic aberrations were significantly reduced due to the extended DOF (see Supplementary Fig 4 and Supplementary Fig 5).

The ability of DeepDOF-SE to resolve various clinically relevant features for samples within the target DOF was evaluated using thin frozen-section tissue slides. We obtained images of human colon, esophagus, and liver slides that were stained with DAPI and Rhodamine B as slides were translated throughout the target DOF of DeepDOF-SE and compared results to a conventional fluorescence microscope. For better visualization, we performed a color space transform using the Beer-Lambert method[28]. Figure 4 compares the images taken with DeepDOF-SE and a conventional microscope. DeepDOF-SE consistently resolves varied cellular morphology within the targeted DOF while images acquired with the conventional microscope images suffered from significant blur when the target is out of focus. This observation is corroborated by the Multi-scale Structure Similarity Index Measure (MS-SSIM[36]) score when using the in-focus image as a reference. The MS-SSIM for images acquired with conventional microscopy quickly drops to as low as 0.39 while DeepDOF-SE images maintain a high MS-SSIM (0.85+) across the 200 μm DOF.

### DeepDOF-SE imaging of fresh tissue without refocusing

We validated the performance of DeepDOF-SE to image fresh resected tissue specimens and images from a porcine kidney specimen (Fig. 5, top row) and a surgical resection of human oral mucosa (Fig. 5, bottom row). Specimens were stained with DAPI and Rhodamine B, then imaged with both the conventional microscope and DeepDOF-SE for comparison. The images were virtually stained using Beer-Lambert method for better visualization.

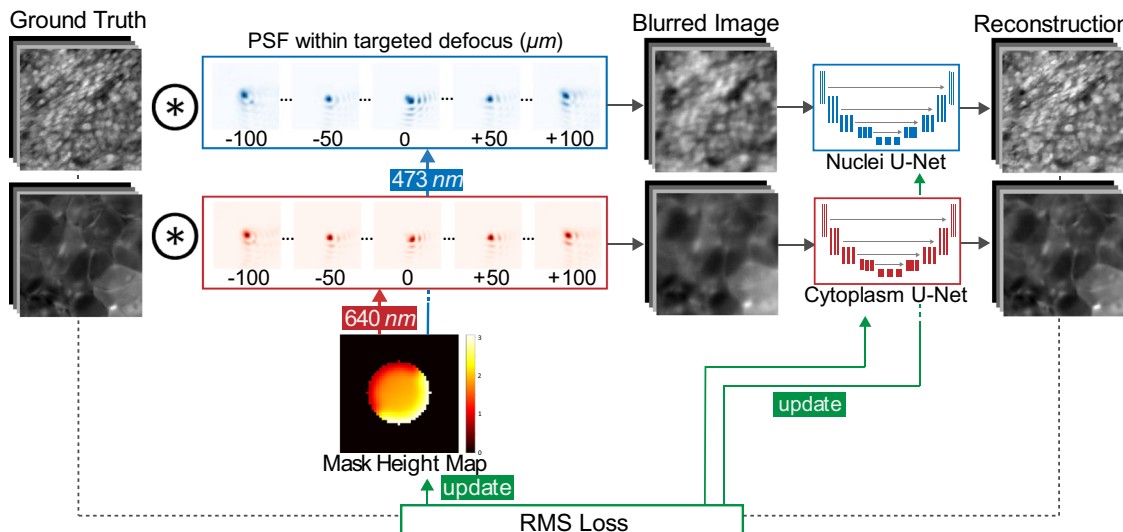

**Fig. 2 | An end-to-end deep learning network to jointly design the imaging optics and image processing for extended depth-of-field imaging in two fluorescence channels.** The end-to-end (E2E) network first simulates the physics-derived image formation of a fluorescence microscope with a learned phase mask and produces simulated blurred images; then the sequential image processing layers consisting of two U-Nets reconstructs in-focus images within the targeted DOF of 200 μm. Both the phase mask design and the U-Net weights are optimized based on the loss between the ground truth images and the corresponding reconstructed images. PSF point spread function, RMS root mean square.

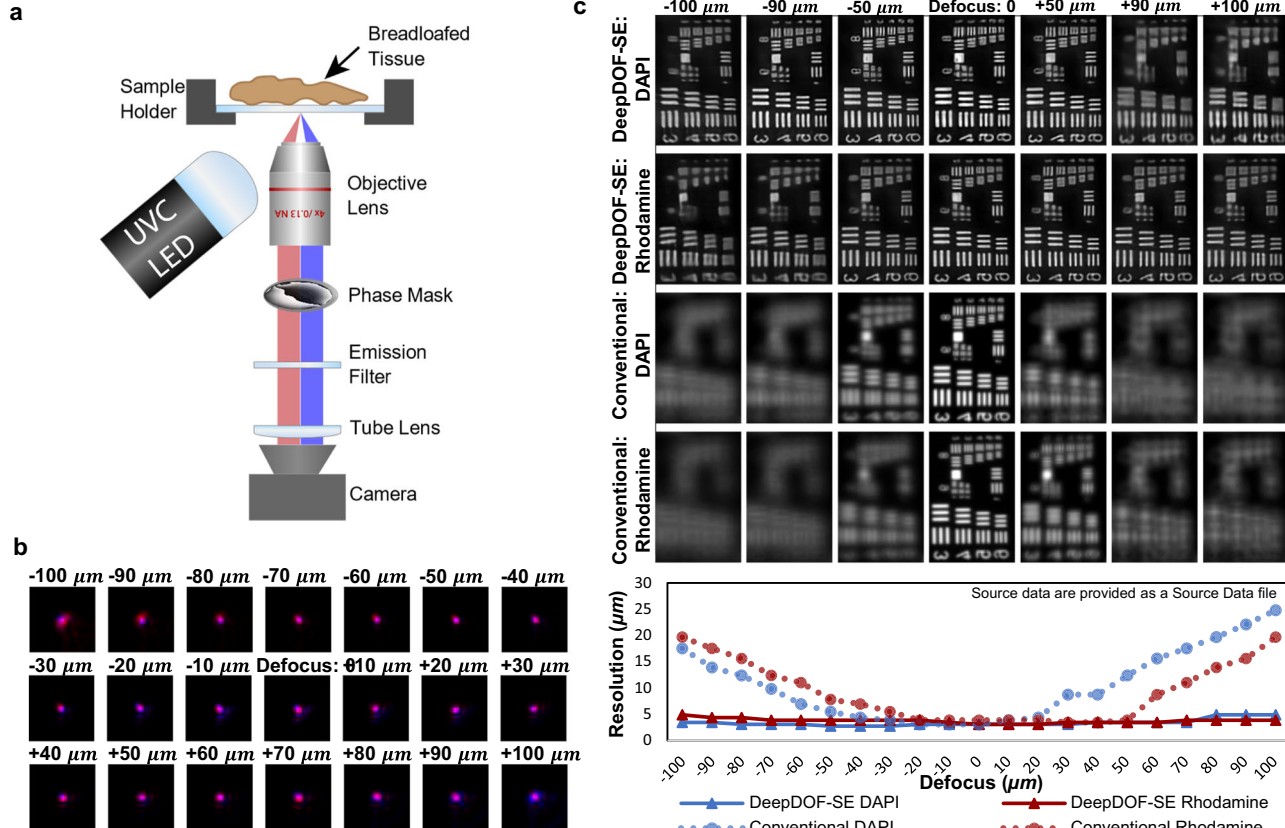

**Fig. 3 | Optical setup and characterization of DeepDOF-SE. a** The optical schematic of DeepDOF-SE. The DeepDOF-SE microscope is built based on a simple fluorescence microscope, with the addition of a deep-learning-enabled phase mask that enables an extended DOF and a UVC LED that enables surface excitation. **b** Experimentally captured point spread functions at 21 discrete depths within the 200 μm target DOF. **c** Experimental resolution characterization of the spatial

resolution of DeepDOF-SE in DAPI and Rhodamine B channels using a USAF 1951 resolution target, in comparison to a conventional fluorescence microscope as the baseline. DeepDOF-SE consistently resolves Group 7, element 5 (2.46 μm line width) or better in both color channels within the target DOF; in addition, DeepDOF-SE exhibits significantly reduced chromatic aberration compared to the conventional microscope.

For each sample, we selected ROIs that appeared in focus (ROIs 4, 5, 9, 10) and out of focus (ROIs 1, 2, 3, 6, 7, 8) in images collected with the conventional microscope. It was challenging to resolve nuclear features in ROIs that were out-of-focus with the conventional microscope. In contrast, nuclei were clearly resolved in all ROIs of images captured with DeepDOF-SE. Similar subcellular features are present in DeepDOF-SE images and in-focus ROIs imaged with the conventional microscope.

## DeepDOF-SE virtual histology via CycleGAN

We model the virtual staining of DeepDOF-SE images as an image-to-image translation that aims to generate images with histology features similar to those in the corresponding standard H&E images. However, it is challenging to acquire image pairs from fresh tissue specimens at the same imaging planes using DeepDOF-SE and standard H&E processing; as a result, we trained the image-to-image mapping network to virtually stain DeepDOF-SE images as part of the CycleGAN architecture (Fig. 6a). Unlike deep-learning networks that perform pixelwise translation, CycleGAN can be effectively trained without paired image sets. As shown in Fig. 6a, the two domains $X$ and $Y$ are defined as DeepDOF-SE images and standard H&E images, respectively. The image mapping in each direction is trained using an adversarial architecture; from domain $X$ to $Y$, for example, the generator $G$ aims to virtually stain DeepDOF-SE images, while the discriminator network $D_Y$ aims to distinguish virtually stained H&E images generated by $G$ from standard H&E images.

Since generative networks can be prone to synthesizing unwanted features[31], we implemented a semi-supervised training procedure (described in "Methods" section) to ensure that both nuclear and cytoplasmic features are accurately translated. In Fig. 6b, c, we validate its ability to preserve clinically important features. To validate Cycle-GAN virtual staining, we used frozen section slides of mouse tongue, which allowed us to acquire co-registered DeepDOF-SE and standard H&E images; the DeepDOF-SE fluorescence images were then virtually stained using the CycleGAN. We applied automated segmentation algorithms[37] to both the CycleGAN H&E and the standard H&E to count the number of nuclei and calculate the average nuclear area (see Supplementary Table 2 and Supplementary Fig 9). Nuclear counts for four selected FOVs are displayed in Fig. 6b. The close agreement between the number of nuclei in CycleGAN stained samples and H&E samples supports the ability of CycleGAN staining to preserve important clinical features. Figure 6c shows CycleGAN stained images and H&E stained images of two fields of view (FOV1 and FOV2). In the first FOV (left panel, Fig. 6c), cross-sectioned muscle fibers are clearly shown with evenly distributed nuclei in the CycleGAN virtual HE image; similar features are observed in the gold standard H&E image. The second FOV (right panel, Fig. 6c) shows the epithelium with underlying lamina propria and muscle fibers; the layered epithelial cell structure and the basement membrane are clearly shown in the CycleGAN virtual H&E image. Importantly, when compared to the gold standard H&E images, the CycleGAN virtual H&E images show co-localized nuclei and cytoplasmic features across the entire FOV, confirming that the

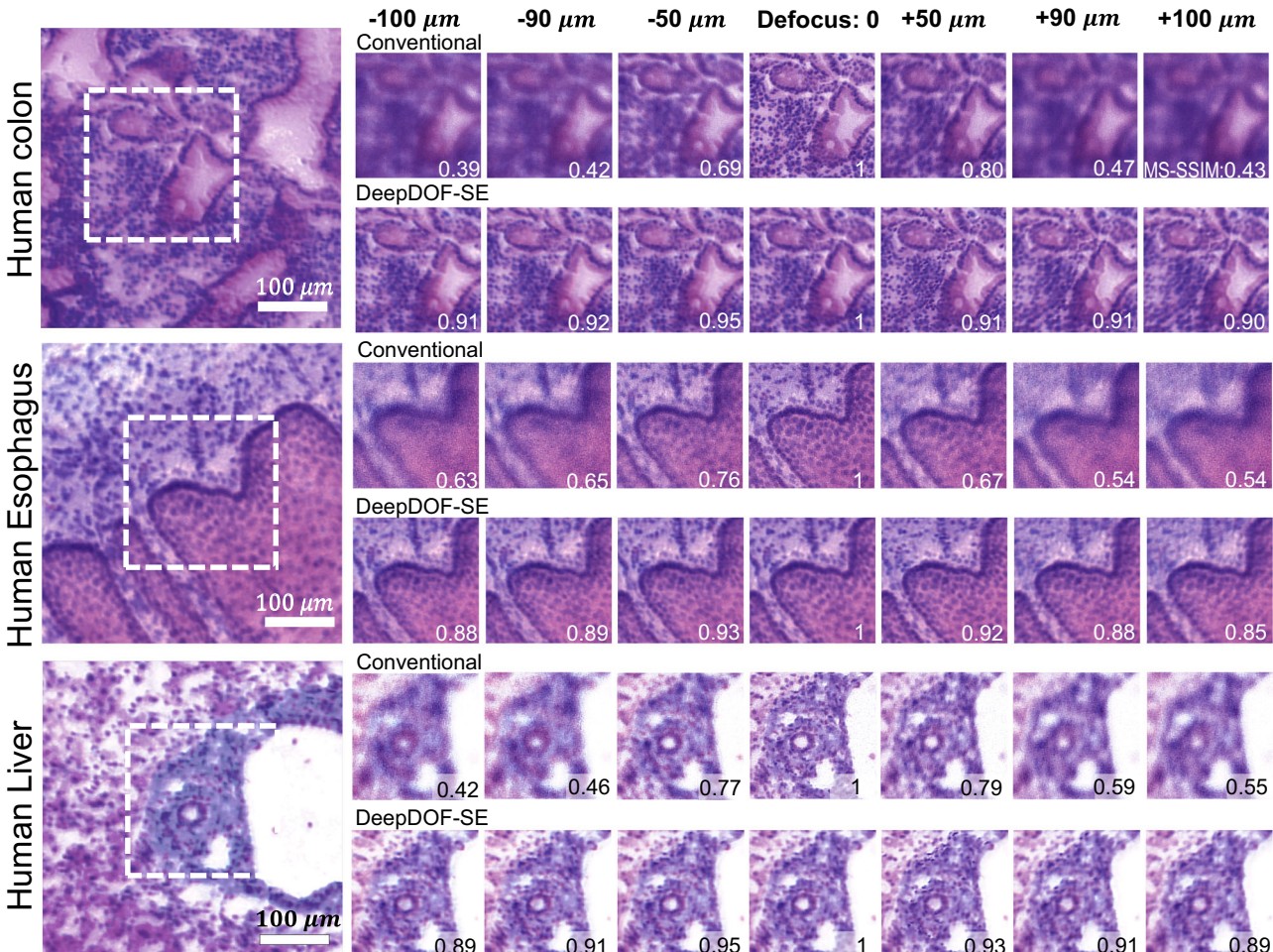

**Fig. 4 | Images of thin (7–10 μm) frozen tissue sections of varied types acquired with DeepDOF-SE and a conventional microscope as the sample is translated axially throughout the target DOF.** All images are virtually stained using the Beer-Lambert method, an analytical color space transform to better visualize the sub-cellular features while preserving defocus artifacts. Virtually stained images from human tissue sections revealed architectural and cellular morphology of colonic crypts lined by intestinal columnar epithelium (top panel), esophagus lined by stratified squamous epithelium (middle panel), and bile duct and portal vein within the portal tract of the liver (bottom panel). In all tissue types, cell nuclei are consistently resolved in images acquired with DeepDOF-SE, while significant defocus blur was observed in images acquired with the conventional microscope. The multiscale structural similarity index MS-SSIM was consistent as the sample was translated throughout the DOF of DeepDOF-SE, while the MS-SSIM dropped rapidly as the sample was translated across the focal plane of the conventional microscope. Scale bars are 100 μm. MS-SSIM multiscale structural similarity index measure.

CycleGAN performs virtual staining while accurately preserving clinically important features. We note that there exist minor color differences between the CycleGAN virtual H&E and standard H&E images, which can be attributed to the known color variations during standard H&E processing[38].

## H&E validation for DeepDOF-SE virtual staining in human oral tissue specimen

We assessed the diagnostic potential of the DeepDOF-SE microscope by comparing DeepDOF-SE images of freshly resected tissue virtually stained using CycleGAN to the gold-standard FFPE H&E scan of the same tissue. Figure 7 shows images of two large specimens from freshly resected head and neck squamous cell carcinoma. Each specimen was transected with a scalpel; fluorescence images (top row), virtually stained images using the CycleGAN from the DeepDOF-SE (middle row) are displayed in Fig. 7. Subsequently, FFPE H&E sections were prepared from the same samples (Fig. 7, bottom row). Representative ROIs from these specimens reveal various histopathologic features in different tissue types and disease status; importantly, matching cellular details between the DeepDOF-SE GAN staining images and FFPE H&E images are observed in these ROIs. We note that

there exist subtle color differences between CycleGAN virtual staining and standard H&E staining.; these differences are quite similar to variations in the intensity of staining that occur from lab to lab and daily within a single lab. Variations in factors such as the age of stains or the precise staining time can lead to intensity variations and overstaining issues in H&E stained slides[39]. Despite these differences, epithelial architecture and cellular detail are clearly discerned in both, providing sufficient diagnostic information for clinical evaluation.

Four selected FOVs from the surface epithelium with underlying connective tissue and skeletal muscle bundles are shown across each of the cross-sectioned specimens in Fig. 7. In the stratified squamous epithelial layers, ROIs 1, 2, 5, 6 show the individual nuclei with clearly visible basal layer with attached basement membranes. Specifically, ROI 1 shows hyperplasia with dysplasia, while ROIs 2, 5, and 6 display hyperkeratosis, evidenced by the increased thickness of the surface keratin. Invasive islands of squamous cell carcinoma characterized by cellular and nuclear atypia and dyskeratosis were noted within the connective tissue underlying the surface epithelium in ROI 7. ROIs 3 and 4 show skeletal muscle bundles below the lamina propria that are sectioned in cross- and longitudinal-directions found in the H&E section were also observed in the DeepDOF-SE images. A large muscular

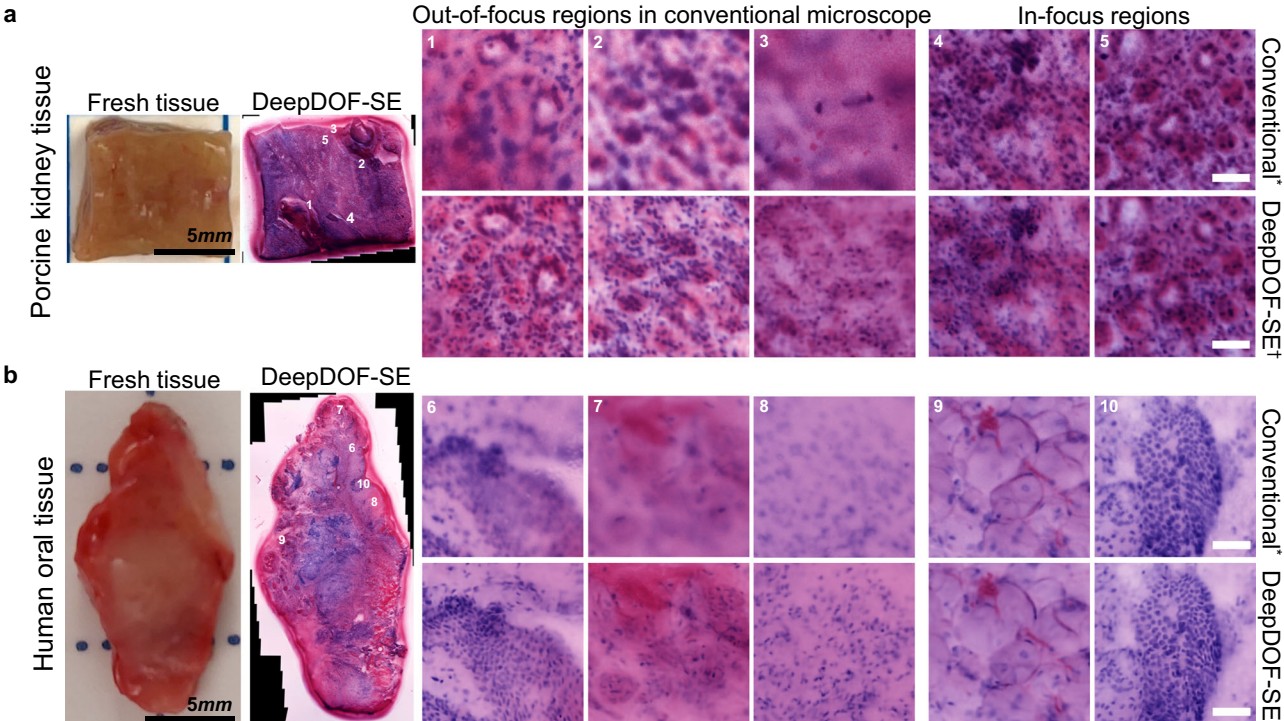

**Fig. 5 | Images of intact fresh tissue of varied types obtained using DeepDOF-SE and a conventional microscope without refocusing.** Conventional*: Conventional microscope (4× 0.13 NA) with 280 nm excitation, with virtual staining using the Beer-Lambert method; DeepDOF-SE†: DeepDOF-SE microscope, with virtual staining using the Beer-Lambert method. **a** Ex vivo porcine kidney sample with five annotated ROIs. Conventional microscope images from ROIs 1–3 are out-of-focus while ROIs 4 and 5 are in focus. Corresponding DeepDOF-SE images are in focus for all ROIs. **b** Ex vivo human tongue resection with five annotated ROIs. Conventional microscopy images from ROIs 6–8 are out-of-focus while ROIs 9–10 are in focus. Corresponding DeepDOF-SE images are in focus for all ROIS. ROI scale bars are 50 μm.

artery that was noted within submucosa of H&E stained section can be also identified in the DeepDOF-SE image in ROI 8. These findings were confirmed by the study pathologist (N.V) through standard histopathology evaluation.

## Discussion

We demonstrated optically sectioned, high resolution and extended depth-of-field imaging of intact tissue samples using DeepDOF-SE, a platform designed using a data-driven approach for slide-free histology. The key components of DeepDOF-SE, including deep UV excitation, end-to-end designed extended DOF, and cycleGAN-based virtual staining, jointly enable rapid and slide-free histological examination of fresh tissue specimens with highly irregular surfaces. We show that DeepDOF-SE images reveal a broad range of diagnostic microscopic features within large areas of tissue cross sections. Moreover, varied types of histological architecture in benign and neoplastic conditions are clearly visualized in the CycleGAN virtually stained H&E images, and histologic findings based on DeepDOF-SE images are confirmed by the gold standard H&E histopathology.

Unlike conventional histopathology that is time-consuming and requires expensive equipment, the DeepDOF-SE platform is low-cost to build (less than $7000, see Supplementary Table 1), requires minimal training to use, and takes less than 10 min to stain and image a 7 cm$^2$ tissue sample (4 min for tissue staining and <5 min for tissue scanning). Compared to conventional pathology requiring mechanical sectioning using a microtome, we employ a simple yet effective optical sectioning approach via deep UV excitation. Conventional histopathologic diagnosis is based on H&E-stained tissue sections and hence, pathologists are accustomed to interpreting H&E stained tissue sections. Using DAPI as the nuclear stain and Rhodamine B as the counter stain, DeepDOF-SE can image cell nuclei and cytoplasmic features. We apply the deep-learning-based CycleGAN to virtually stain the all-in-focus fluorescence images. The resulting virtual H&E images revealed diagnostic histopathology matching the corresponding standard slide-based H&E images. While it is challenging to achieve serial sectioning with DeepDOF-SE in cases where diagnosis on the surface is equivocal, it is possible to rapidly scan the opposite side of a 4 mm tissue slice using DeepDOF-SE. Alternatively, the slice can be further cut with a scalpel in 2–3 mm steps before scanning again.

Recent advances in microscopy have enabled a range of fast, nondestructive and slide-free histological imaging technologies, such as MUSE, FF-OCT, the light-sheet microscopy, and photo-acoustic microscopy[13–15,18]. Other methods of extending DOF exist, primarily achieved via axial focus stacking. These methods leverage different mechanisms and instrumentation, such as a deformable mirror, a varifocal lens, or a digital micromirror device[40–42]. Compared to these modalities that usually require complicated optical configurations, DeepDOF-SE leverages a simple optical modulation element with deep learning to substantially augment the performance of a fluorescence microscope for high-throughput, single-shot histology imaging. As a result, the DeepDOF-SE platform can be readily built using a modular design approach at a low cost; all of its key components, including the external deep UV LED, the phase mask, the fluorescence filter, the sample stage and computing hardware, can be seamlessly integrated into a simple microscope system with minimal optical and hardware modification. Moreover, the fast and slide-free tissue preparation requires minimal training and does not interrupt standard-of-care procedures, making the technology suitable for broad dissemination in resource-constrained settings. Our initial clinical assessment of DeepDOF-SE demonstrates its capability to rapidly provide histological information of fresh surgical specimens in Fig. 7, including those needed for the diagnosis of precancer and cancer, such as

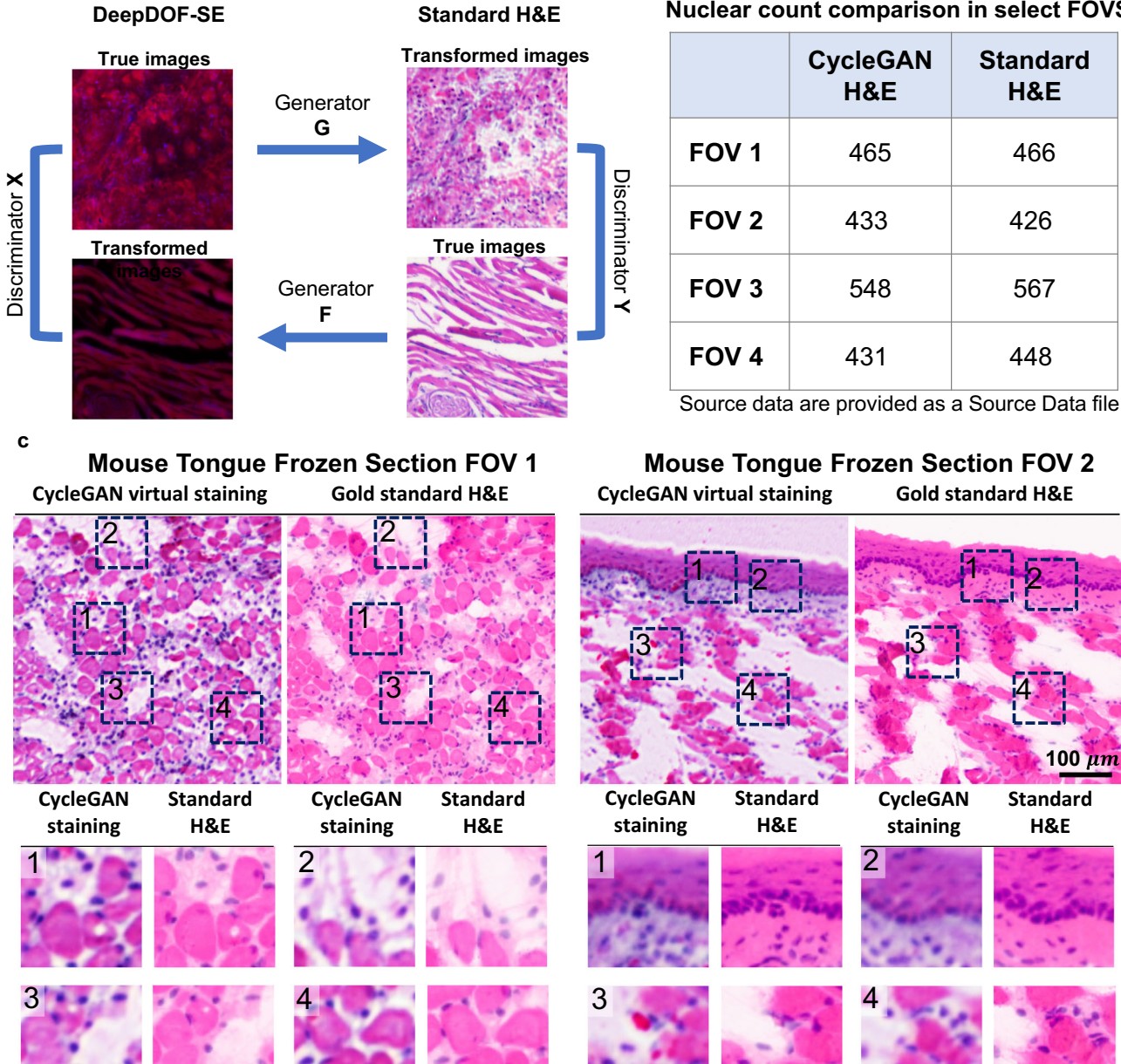

**Fig. 6 | Schematic and validation of CycleGAN virtual staining in DeepDOF-SE.**
**a** Design of a CycleGAN architecture to virtually stain DeepDOF-SE images. The image-to-image mapping network G is trained to virtually stain DeepDOF-SE images using a semi-supervised training strategy without paired DeepDOF-SE images and standard H&E images. **b** Quantitative validation of CycleGAN by comparing nuclear count using an automated software for four field of views between cycleGAN stained H&E and standard H&E of a frozen section of mouse tongue frozen section. Each FOV is 504 × 504 μm. Two fields of view (FOV 1 and FOV 2) are shown in (**c**). **c** CycleGAN stained and standard H&E images from two different FOVs of the frozen section of mouse tongue; insets demonstrate accurate staining of nuclear and cytoplasmic features. Scale bar: 100 μm.

architectural abnormalities, pleomorphism, and abnormal nuclear morphology and increased nuclear-to-cytoplasmic ratio[43]. In cases where a higher resolution is desired, our approach can serve as a rapid triage tool to identify suspicious regions for further examination at a higher magnification. Based on our results, further evaluation with a larger sample size is warranted. In a larger study, using standard H&E as a baseline, we will establish diagnostic criteria based on DeepDOF-SE images, and we will also refine the criteria since it was previously shown that nuclear count in optically sectioned fluorescence images using 280 nm excitation is slightly elevated than conventional H&E[18].

To facilitate its evaluation in a clinical setting, we will enclose the system in a compact housing. In addition, the imaging throughput can be further improved by incorporating a high-sensitivity sensor, higher levels of illumination and faster sample scanning motors.

The DeepDOF-SE platform leverages two deep learning networks in its system design and data processing pipeline, and we employed different training strategies based on the nature of their tasks. The end-to-end extended DOF network aims to simulate physics-informed image formation and reconstruction that are insensitive to image content, and therefore, a data-agnostic approach was used for training.

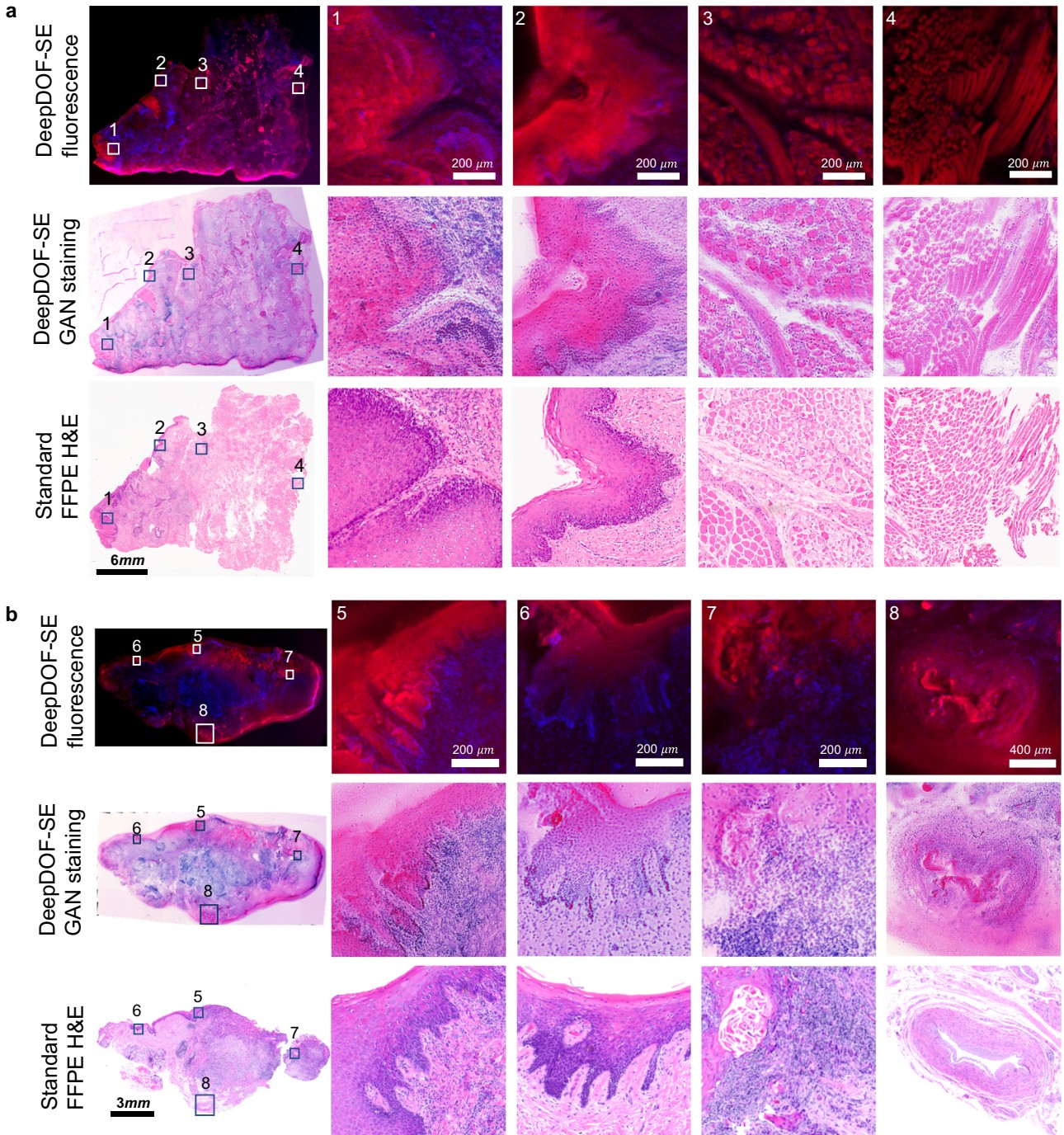

**Fig. 7 | DeepDOF-SE images of oral surgical specimens and corresponding standard H&E images. a, b** DeepDOF-SE fluorescence images (top row), DeepDOF-SE CycleGAN virtual stained images (middle row), and conventional FFPE H&E images (bottom row) of ex vivo human tongue resection. DeepDOF-SE visualizes a broad range of important diagnostic features that are consistent with the gold standard H&E. ROI 1: Epithelial hyperplasia with dysplasia. ROI 2: Epithelial hyperkeratosis and hyperplasia with dysplasia. ROIs 3 and 4: Skeletal muscle bundles. ROIs 5 and 6: Epithelial hyperkeratosis and hyperplasia. ROI 7: Invasive squamous cell carcinoma with dyskeratosis. ROI 8: Muscular artery. FFPE formalin-fixed and paraffin-embedded, H&E hematoxylin and eosin.

In contrast, since the CycleGAN virtual staining network is designed to perform domain-wise image translation, we confine the training and validation scope using images from the tongue in our current study. Specifically, in the extended DOF network, we used an eclectic training dataset containing various features ranging from multiple types of FFPE H&E images to natural scenes; during validation and testing, fluorescence images of different tissue types are reconstructed. This variability can help the model become more robust and adaptable to different types of inputs during inference, allowing it to generalize to a wider range of applications. Data fusion and dataset merging has also been used in other domains of imaging[44–46]. In contrast, the cycleGAN was trained and validated with images from oral tissue surgeries in a clinical study and frozen slides of mouse tongue. While it faithfully translates the DeepDOF-SE fluorescence images of oral tissue to standard H&E appearance, further data collection and clinical evaluation are needed to extend the GAN-based virtual staining to other tissue types. We also note that adipose cells appear intact in DeepDOF-SE images of fresh tissue (Supplementary Fig 13), while they show a

network of thin cell membranes with clear lumens in standard H&E. This is expected since the cytoplasmic lipids within the adipocytes are removed during tissue dehydration using different concentrations of alcohol.

In conclusion, we developed a deep-learning enabled DeepDOF-SE platform that enhanced the ability of conventional microscopy to image intact, fresh tissue specimens without the need for extensive sample preparation, and we validated its performance to provide diagnostic information in oral surgical resections as confirmed by standard slide-based histopathology. As a fast, easy-to-use, and inexpensive alternative to standard histopathology, DeepDOF-SE will be further evaluated clinically, especially for intraoperative tumor-margin assessment and for use in under-resourced areas that lack access to standard or frozen section histopathology.

## Methods

### Ethical statement
Our research involves an ex vivo protocol where consenting patients undergoing surgery for oral cancer resection were enrolled at the University of Texas MD Anderson Cancer Center. The study was approved by the Institutional Review Boards at the University of Texas MD Anderson Cancer Center and Rice University.

### DeepDOF-SE microscope setup
As shown in Fig. 3, The DeepDOF-SE microscope is built using a dual-channel fluorescence microscope with UV surface excitation[18] and the addition of a deep-learning optimized phase mask. The UV LED (Thorlabs M275L4), coupled with a condenser and focusing lens, is pointed at an oblique angle to the sapphire sample window (Knight-Optical, WHF5053), illuminating the sample uniformly from beneath. Fluorescence emission from the vital-dye-stained tissue sample is collected by an Olympus 4× objective (RMS4x-PF, 0.13 NA), modulated by the phase mask, and then an image is relayed by a f150 mm tube lens (Thorlabs AC254-150-A) onto a 20-megapixel color CMOS camera (Tucsen FL20). A dual-bandpass filter (Chroma 59003 m, 460/630 nm) is used for collecting fluorescence from the Rhodamine B and DAPI channels simultaneously.

For convenient placement of large surgical specimens, the DeepDOF-SE has an open-top sample stage with a circular imaging window 50 mm in diameter. Rapid scanning is enabled by two motorized linear stages (Zaber X-LHM100A). With the designed 3.3× magnification, the FL20 camera provides a $3.9 \times 2.6 \text{ mm}^2$ field-of-view per frame. To ensure sufficient overlap between frames for field of view stitching, we chose 2 mm for stage step motion and scanned 40 frames per minute. The scanning process is controlled and automated using a custom LabVIEW GUI. Briefly, using the GUI, the scanning region is first defined with user input, and images are then acquired and saved sequentially using scanning coordinates automatically calculated based on the scanning range and stage step size.

### Optical and digital layer of DeepDOF-SE
Previously, we developed a physics-informed deep-learning network to jointly optimize the imaging optics and image reconstruction for EDOF imaging in a single fluorescence channel[23]. In this work, we employ a similar architecture to enable EDOF imaging in a dual-channel fluorescence microscope. Overall, the end-to-end network consists of an optical layer to optimize the imaging optics and a digital layer to optimize the image reconstruction.

**Optical layer**. The first layer of the end-to-end extended DOF network parameterizes the design of a phase mask and simulates image formation of a dual-channel fluorescence microscope from the specimen to the sensor, with its wavefront at the pupil plane modulated by the phase mask. In the current work, we design the deep learning

network for two fluorescence channels centered at 473 nm and 640 nm, corresponding to emission of DAPI and Rhodamine B, respectively.

The image formation is simulated based on Fourier optics[47]. Briefly, in each fluorescence channel, an image $I_\lambda(x_2,y_2)$ formed by the microscope is the result of scene $I_0(x,y)$ convolved with the point spread function (PSF) at the given wavelength $\lambda$ summed across depth $z$.

$$I_\lambda(x_2,y_2) = \sum_z I_0(x,y;z) \otimes PSF_\lambda(x_2,y_2;z) \tag{1}$$

The PSF is the squared magnitude of the Fourier transform of the pupil function $P_\lambda(x_1,y_1;z)$

$$PSF_\lambda(x_2,y_2;z) = \left| \mathcal{F}\{P_{\lambda(x_1,x_2;z)}\} \right|^2 \tag{2}$$

With the amplitude of the pupil function fixed, the phase component of the pupil function ($\Phi$) encodes the defocus blur $\Phi^{DF}$ and the depth-independent mask modulation $\Phi^M$.

$$\Phi_\lambda(x_1,y_1;z) = \Phi_\lambda^{DF}(x_1,y_1;z) + \Phi_\lambda^M(x_1,y_1) \tag{3}$$

In the equation above, the mask modulation term $\Phi^M$ is modulated by the height map of the phase mask, which is parameterized using the first 55 Zernike basis in the first layer of the end-to-end optimization network. In addition, the defocus phase is modeled as

$$\Phi_\lambda^{DF}(x_1,y_1;z) = \frac{2\pi}{\lambda} W_m \frac{x_1^2 + x_2^2}{R^2} \tag{4}$$

where $R$ is the radius of the pupil and $W_m = \frac{R^2}{2} * \frac{(z_0-z)}{z_0^2}$ is the maximum path-length error at the edge of the pupil due to defocus where $z$ and $z_0$ are the defocused imaging depth and in-focus depth respectively.

For a given scene, we simulate the final sensor image from two wavelengths corresponding to the two fluorescence channels, and 21 discrete depths evenly discretized in the targeted DOF range of 200 μm. This corresponds to $\frac{2\pi}{\lambda} W_m$ ranges of $[-8.73, +8.73]$ at 473 nm and $[-11.88, +11.88]$ at 640 nm. We also approximated the sensor noise by adding a Gaussian read noise with a standard deviation of 0.01 in the range of [0, 1].

**Digital layer**. Sensor images from different defocus in two fluorescence channels are further processed using the image reconstruction layers to recover in-focus images of the specimen. As shown in Fig. 2, the digital layer consists of two deep neural networks, and we used a U-Net architecture described in Jin et al.[23].

### Network training details
**Dataset**. To ensure the system is capable of imaging a wide variety of features, we train the network with a large dataset that contains a broad range of imaging features[23]. Specifically, the dataset contains 600 high-resolution proflavine-stained oral cancer resections, 600 histopathology images from Cancer Genome Atlas Center FFPE slides, and 600 natural images from the IRNIA holiday dataset (each $1000 \times 1000$ pixels, gray scale)[48]. While these images have diverse features, they are all in gray scale and cannot be directly used to train DeepDOF-SE, which generates color images. Natural RGB images are also not suitable because the color images captured by fluorescence microscopes contain different information in each color channel. Instead of collecting a new color dataset, which is costly and time-consuming, we randomly combined two different images from the DeepDOF dataset for the DAPI channel and the Rhodamine B channel

as input during training; this provides effective training while eliminating cross-talks between the two fluorescence channels.

The 1800 images in the DeepDOF dataset were randomly separated into training, validation, and testing sets. To increase data variability, the images were augmented with random cropping (from $256 \times 256$ to $326 \times 326$ pixels), rotation, flipping, and brightness adjustment. Since the dataset contains a rich library of features including both histopathological and other features in nature scenes, it is broadly applicable to training image reconstruction pipelines using different microscope objectives with proper rescaling.

**Implementation.** The network was implemented using the TensorFlow package and optimized using Adam[49]. The learning rate was chosen empirically at 1e-9 for the optical layer and 1e−4 for the digital layer. We trained the network in two steps. In the first step, we fixed the optical layer to be the cubic mask and only trained the U-Net. In the second step, we jointly trained the optical and digital layer. For both steps, convergence occurred at around 30,000–40,000 iterations.

### Microscope calibration and network fine-tuning

To account for the difference between the simulated PSF and the experimental PSF during system implementation, we performed a one-time calibration. A monolayer of 1 um fluorescent beads (Invitrogen T7282, TetraSpeck microspheres, diluted to $10^5$/mL) was imaged as a calibration target, and we adjusted the right-angle mirror behind the objective and a micrometer tilt stage (#66-551, Edmund Optics) installed on the sample stage to achieve uniform focus across the sample imaging window.

To capture the PSFs in the two fluorescence channels, we used fluorescent TetraSpeck beads stained with four fluorescent dyes at 360/430 nm (blue), 505/515 nm (green), 560/580 nm (orange), and 660/680 nm (dark red). The beads were illuminated using a 365 nm LED (Thorlabs M365LP1) for better excitation, and PSFs were measured at 31 depths at 10 um intervals. At each depth, we averaged temporally over five frames and performed background subtraction to reduce noise.

We selected 21 depths for the target DOF for network fine-tuning. When retraining the network, the optical layer was fixed and the experimentally captured PSF was used to fine-tune the network.

### Resolution characterization

The resolution of DeepDOF-SE was characterized by imaging a US Air Force 1951 resolution target with an added fluorescent background. Illumination was provided by a 405 nm LED, and we performed frame averaging and background subtraction to enhance the signal-to-noise ratio.

### Tissue processing and imaging

**Human surgical sample.** Fresh surgical cancer resections from the oral cavity were imaged to evaluate the imaging performance of DeepDOF-SE. In our ex vivo protocol, consenting patients undergoing surgery for oral cancer resection were enrolled. The excised specimen was first assessed by an expert pathologist and sliced into 3–4-mm-thick slices with a standard scalpel. Selected slices were processed for standard frozen-section pathology. The remaining slices were cleaned with phosphate-buffered saline (PBS, Sigma-Aldrich P4417, pH 7.2–7.6, isotonic) to remove residuals such as mucus and blood, stained with DAPI (Sigma-Aldrich MBD0015, diluted with PBS, 500 μg/mL) for 2 min and Rhodamine B (Sigma-Aldrich 83689, dissolved in PBS, 500 μg/mL) for 2 min, and excessive stain was rinsed off with PBS. The tissue was then imaged using the DeepDOF-SE microscope. The raw frames were processed with the DeepDOF-SE networks and stitched using Image Composite Editor (Microsoft, discontinued and other stitching

software can be applicable). Post-imaging, the specimens were processed through FFPE histopathology at University of Texas MD Anderson Cancer Center, and the slides were imaged using a slide scanner to provide the standard H&E images. The study was approved by the Institutional Review Boards at the University of Texas MD Anderson Cancer Center and Rice University.

**Porcine tissue.** Freshly resected ex vivo porcine samples were obtained from an abattoir. The tissue was cut with a scalpel, cleaned with PBS to remove residuals such as mucus and blood, and then stained with DAPI (500 μg/mL) for 2 min and Rhodamine B (500 μg/mL) for 2 min. Excessive stain was rinsed off with PBS, and the tissue was imaged using DeepDOF-SE and a conventional MUSE microscope with the same standard objective.

**Frozen-section slides.** Frozen-section tissue slides (Zyagen, Inc) were fixed in buffered acetone (60% v/v) for 20 min and rinsed in PBS twice for five minutes each. Slides were then stained with DAPI (500 μg/mL) for 2 min and Rhodamine B (500 μg/mL) for 2 min, and excessive stain was rinsed off with PBS. The stained slide was imaged with DeepDOF-SE without a coverslip on the sapphire window, with the tissue side facing downward. Since glass slides have autofluorescence, we subtract the background signal before any downstream processing. For the cycleGAN validation study, the imaged frozen section slides were sent to University of Texas MD Anderson Cancer Center for standard H&E staining.

**Statistics & reproducibility.** In this study, no statistical method was used to predetermine sample size. For Figs. 4 and 5 of the main text, the samples are imaged once with the proposed DeepDOF-SE and once with the conventional baseline. For Figs. 6 and 7, the samples are imaged once with the proposed DeepDOF-SE.

### Beer-Lambert-law-based virtual H&E

We used a Beer-Lambert-law-based method to assist visualization of DeepDOF-SE images in a color space similar to H&E staining[28]; since it is an analytical method, it preserves both in- and out-of-focus features in DeepDOF-SE images. In this virtual staining method, the transmission $T$ of a wavelength $\lambda$ through a specimen containing $N$ absorbing dyes can be represented as

$$T_\lambda = \exp\left(-\sum_{i=1}^{N} \sigma_{\lambda,i} c_i\right) \quad (5)$$

Where $\sigma_{\lambda,i}$ is the wavelength-dependent attenuation for the $i$-th dye and $c_i$ is the thickness integrated concentration of the $i$-th dye per area on the slide. In the case of a digital image, the transmission $T_M$ of $M$-th color channel can be written as

$$T_M = \exp\left(-\sum_{i=1}^{N} \beta_{M,i} I_i k\right) \quad (6)$$

Where $\beta_{M,i}$ is the attenuation of the $i$-th dye integrated over the spectral range of the $M$-th channel, $I_i$ is the intensity image for $i$-th dye, and $k$ is an arbitrary scaling constant that accounts for detector sensitivity etc. In the case of mapping to H&E staining in RGB space, the expression for each channel is as follows:

$$R = \exp(-\beta_{hematoxylin,red} I_{nucleistain} k) \exp(-\beta_{eosin,red} I_{counterstain} k) \quad (7)$$

$$G = \exp(-\beta_{hematoxylin,green} I_{nucleistain} k) \exp(-\beta_{eosin,green} I_{counterstain} k) \quad (8)$$

$$B = \exp(-\beta_{hematoxylin,blue} I_{nucleistain} k) \exp(-\beta_{eosin,blue} I_{counterstain} k) \quad (9)$$

We empirically chose $k$ to be 2.5 for images of range [0, 255]. The values of the $\beta$s are described in Giacomelli et al.[28].

## CycleGAN-based virtual H&E

**Network architecture.** The domain-wise image translation from DeepDOF-SE images (domain $X$) to standard H&E images (domain $Y$) was trained using CycleGAN, a network architecture capable of unpaired image-to-image translation[32]. Briefly, the network consists of two generators, $G$ that maps DeepDOF-SE images ($X$) to H&E images ($Y$) and $F$ that maps H&E images ($Y$) to DeepDOF-SE images ($X$). For each generator, a discriminator network is tasked to distinguish images synthesized by the generators from the ground truth image set ($D_X$ for $F$ and $D_Y$ for $G$). The generator networks are 9-block ResNets and the discriminator networks are $70 \times 70$ PatchGANs; instance normalization is implemented in all networks.

We aim to train the CycleGAN so that the generators perform realistic color and texture translation while accurately preserving nuclear and counterstain features. To achieve this goal without accurately co-registered ground truth images in domains $X$ and $Y$, we adopted a two-step semi-supervised training strategy (Supplementary Fig. 7). In step 1, to pretrain the generators for color translation with co-registered features, we synthesized a paired training set consisting of DeepDOF-SE images ($X$) and the corresponding Beer-Lambert-based false-colored H&E images ($\hat{X}$). During this step, the generators were trained to perform the color mapping, while the feature correspondence (e.g., nuclei in DAPI channel of DeepDOF-SE images to nuclei in eosin channel in H&E images) between the two domains is preserved. In step 2, we used unpaired DeepDOF-SE images ($X$) and standard H&E images ($Y$) to retrain the CycleGAN. Compared to a CycleGAN directly trained with a dataset of unpaired images in an unsupervised manner, the semi-supervised training ensures that both nuclear and contextual features are accurately preserved (Supplementary Fig. 8).

**Loss function.** The objective used to train the GAN consists of loss terms for the generator and the discriminator in each mapping direction, and a cycle consistency loss for the two generators[50]. More specifically, the GAN losses for the generators and discriminators are:

$$\mathcal{L}_{GAN}(G,X,Y) = \mathbb{E}_{x \sim p_{data}(x)}\left[\left(D_Y(G(x)) - 1\right)^2\right] \tag{10}$$

$$\mathcal{L}_{GAN}(D_Y,X,Y) = \mathbb{E}_{x \sim p_{data}(x)}\left[\left(D_Y(G(x))\right)^2\right] + \mathbb{E}_{y \sim p_{data}(y)}\left[\left(D_Y(y) - 1\right)^2\right] \tag{11}$$

$$\mathcal{L}_{GAN}(F,X,Y) = \mathbb{E}_{y \sim p_{data}(y)}\left[\left(D_X(F(y)) - 1\right)^2\right] \tag{12}$$

$$\mathcal{L}_{GAN}(D_X,X,Y) = \mathbb{E}_{y \sim p_{data}(y)}\left[\left(D_X(F(y))\right)^2\right] + \mathbb{E}_{x \sim p_{data}(x)}\left[\left(D_X(x) - 1\right)^2\right] \tag{13}$$

The cycle consistency loss, which ensures that the synthesized images can be mapped back to the original ground truth images through a cycle is as follows:

$$\mathcal{L}_{cyc}(G,F) = \mathbb{E}_{x \sim p_{data}(x)}\left[||F(G(x)) - x||_1\right] + \mathbb{E}_{y \sim p_{data}(y)}\left[||G(F(y)) - y||_1\right] \tag{14}$$

The cycle consistency loss is combined with the GAN losses, and as a result, the total losses for the two generators are:

$$\mathcal{L}_{total}(G,X,Y) = \mathcal{L}_{GAN}(G,X,Y) + \lambda_1 \mathcal{L}_{cyc}(G,F) \tag{15}$$

$$\mathcal{L}_{total}(F,X,Y) = \mathcal{L}_{GAN}(F,X,Y) + \lambda_1 \mathcal{L}_{cyc}(G,F) \tag{16}$$

Note that in our training step 1, the standard H&E image domain $Y$ is replaced with the Beer-Lambert-based false-colored H&E image domain ($\hat{X}$).

**Dataset.** The CycleGAN was trained using images of resected surgical tissue from human oral cavity (described in "Tissue processing and imaging"). The training dataset consists of an unpaired image dataset of 604 DeepDOF-SE images and 604 standard H&E images from the same tissue specimen. The standard H&E scans were scaled to match the DeepDOF-SE images, and a patch size of $512 \times 512$ pixels was used. For training step 1, we also performed Beer-Lambert-law-based color mapping to generate paired DeepDOF-SE and false-colored images.

Once trained, we evaluated the trained CycleGAN (specifically, the generator $G$) for mapping DeepDOF-SE images to standard H&E images. First, we validated its performance to accurately map nuclear and cytoplasmic features between the two domains. Since it is challenging to acquire paired DeepDOF-SE and standard H&E images with co-registered features from fresh tissue specimens, we used frozen tissue slides of mouse tongue as a target. We first obtained DeepDOF-SE images of the frozen slides, which were then submitted for standard H&E processing and scanning; the H&E images were aligned to the DeepDOF-SE images through SURF feature matching[51] to generate co-registered image pairs for algorithm validation. Once the algorithm is validated with frozen slide images, we further evaluated its performance in virtually staining DeepDOF-SE images of fresh tissue specimens.

**Training details.** The network was implemented using the TensorFlow package and optimized using Adam[49]. In both steps, the CycleGAN was trained for 5 epochs, with the learning rate empirically chosen at 2e−04.

## Reporting summary

Further information on research design is available in the Nature Portfolio Reporting Summary linked to this article.

## Data availability

The training, validation, and testing datasets for the extended depth-of-field network and the imaging data underlying the figures are available at https://zenodo.org/records/10674605. The source data underlying the graphs in this work are provided in the file "DeepDOF-SE_source_data.xlsx". The training data for the virtual staining cycle-GAN are available under restricted access due to protocol restriction, access can be obtained by contacting the corresponding authors with material transfer agreements. Source data are provided with this paper.

## Code availability

The code used in this study is available on GitHub: https://github.com/MJ2695/DeepDOF-SE/.

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

## Acknowledgements

L.J., Y.T., J.B.C., M.T.T., R.R.R.-K. and A.V. are supported by NSF-1730574. L.J., Y.T., J.B.C., H.B., M.D.W., N.V., A.M.G., R.R.R.-K. and A.V. are supported by NIDCR-R01DE032051. H&E processing was performed by the MD Anderson Research Histology Core Laboratory, which is supported by NCI-CA16672. Additionally, J.T.R. and A.V. are sponsored by the Defense Advanced Research Projects Agency (DARPA) through Contracts No. N66001-17-C-4012 and N66001-19-C-4020 through the Naval Information Warfare Center. The content of the information does not necessarily reflect the position or the policy of the Government, and no official endorsement should be inferred. The results shown here are in part based upon data generated by the TCGA Research Network: https://www.cancer.gov/tcga. The authors would like to thank Cheima Hicheri and Donna Turner for their assistance on frozen slide processing.

## Author contributions

L.J., Y.T., A.M.G., R.R.R.-K. and A.V. designed the end-to-end framework of DeepDOF-SE. L.J., Y.T., J.B.C., M.T.T., J.T.R. and X.Z. developed the staining protocol and hardware prototype of DeepDOF-SE. L.J., Y.T., M.T.T., H.B., M.D.W., N.V., A.M.G., R.R.R.-K. and A.V. led the experiments. L.J., Y.T., J.B.C., M.T.T., M.D.W., A.M.G., N.V., R.R.R.-K. and A.V. analyzed the data. L.J., Y.T., N.V., R.R.R.-K. and A.V. wrote the manuscript. R.R.R.-K. and A.V. supervised the project.

## Competing interests

The authors declare no competing interests.
