## [Peer Review File · Nature Communications]

Reviewers' Comments:

Reviewer #1:

Remarks to the Author:

This is an important evolution for slide-free microscopy, and the combination of low cost and fast performance suggests that this could be widely adopted. My comments are not really sorted in terms of importance.

I recommend acceptance, after attention to some issues. The images were acquired with a 4X, .13 NA objective. This makes capture of high-resolution details difficult--it would be helpful to hear what this implies for the proposed use case, as well as others.

The CycleGAN was only trained on the same specimen in both DeepDOF-SE mode and H&E mode. This certainly encourage the model to perform well, but it would be instructive to see how it performs on other specimens of the same general type.

There are some differences between DeepDOF-SE mode and H&E. One is illustrated in the main text but not discussed--I refer to the images shown in 7b in which the epidermis and dermis are quite different in appearance between DeepDOF-SE mode and H&E. Not fatal, but worth commenting on. Secondly, and more strikingly, the performance on fat (adipose) tissue with DeepDOF-SE mode is very different than in H&E--with fat showing up in bright red whereas it is clear (gone) in the H&E. This is an important performance difference that could have real implications. I strongly suggest that this example be moved to the main text and discussed. There should be technical solutions if this performance proves problematic.

Image 1B fluorescence panels could be brightened. Very hard to see properly.

What is the apparent depth of capture? In the dermis there seem to be about twice the number of cells that what shows up in the H&E equivalent. Worth a sentence or two.

The Supp. section seemed to imply that the DOF approach can work with RGB signals, not just 2-color, but it is not made clear what the story is or could be. Please explain more clearly.

in focus: needs hyphen

Finetune: needs hyphen

ex vivo: needs hyphen

space needed between wavelength and nm, so 405 nm

Reviewer #2:

Remarks to the Author:

The work presented in the manuscript proposes a solution to the histopathological study of tissues obtained by resection during the development of surgery, avoiding the long times of histochemical preparation of standard procedures and the need for expensive infrastructures.

The solution proposed by the authors to achieve this purpose consists of a bi-channel fluorescence microscope with UV excitation that allows the observation of in situ samples obtained by scalpel resection and staining (DAPI and rhodamine B) in a rapid process of a few minutes. The image obtained is processed to obtain EDOF by a model based on a neural network trained from images with depth information, relying on a phase mask. This avoids the need to refocus the sample in a range of 200 microns. This range guarantees that all surface irregularities obtained by manual scalpel resection are correctly focused on the final image.

To facilitate the histopathological interpretation of the florescence images obtained, a virtual H&E staining method based on generative adversarial neural networks with cycle consistency (CycleGAN) is proposed. The proposed method consists of a semi-supervised training in two steps. In the first of them, the model is trained in a supervised way with a set of paired images obtained by applying analytical virtual staining (method based on the Beer-Lambert law) to the images of

fluorescence. In the second step the model weights are fine-tuned through unsupervised training replacing the virtual staining images based on the Beer-Lambert law of the previous step with unpaired images of real standard H&E staining.

Therefore, the work presents a platform capable of performing slide-free histological studies suitable for rapid histopathological evaluation in intraoperative contexts or with limited resources based on bi channel fluorescence microscopy with superficial UV excitation. The platform is capable of scanning 1.6 cm²/min, with EDOF of 200 microns while preserving sub-cellular resolution.

The two most relevant aspects of the work presented refer to the processing of images obtained by UV excitation fluorescence microscopy (MUSE) are:

1. application of the EDOF method, proposed by the authors in the previous work Jin et al. (2020), to a bi channel fluorescence image obtained with DAPI and rhodamine B staining --in the previous work, the image was obtained by conventional single-channel fluorescence microscopy--.

2. H&E virtual staining of fluorescence images using a model based on generative adversarial neural networks with cycle consistency trained in two steps.

Regarding the two previous aspects, the work presents some important weaknesses. These can be summarized in that the manuscript should highlight the main contributions of the work presented and how these represent an advantage over the state of the art of the matter. More specifically:

- Along with the proposed EDOF method, the authors should highlight what are the new contributions with respect to the previous published work (DOI: 10.1073/pnas.2013571117), beyond its application to a bi channel image. In addition, to judge the relevance of the results obtained it would be very enlightening to add a comparison of the alternative final performance with some other EDOF methods (e.g., z-stacking).

- The virtual staining method should be accompanied by a discussion of related works that allow judging the relevance of contributions in this area. In addition, to justify the relevance of the proposed method, it is necessary its quantitative comparison with the result obtained with the virtual staining analytical method based on the Beer-Lambert law.

Both the most important weaknesses mentioned previously, as well as the less important comments, have been contextualized as comments in the files (PDF) provided by the authors.

After careful review of the manuscript and the supplementary material, I think that both require in-depth revision to address the issues that I have identified as particularly important. However, I do not want my opinions to overlook my sincere respect for the work done by the authors and the careful writing of the manuscript that facilitates its reading and understanding.

Reviewer #3:

Remarks to the Author:

In this manuscript entitled "DeepDOF-SE: affordable deep-learning microscopy platform for slide-free histology" by Jin et al., authors describe a technology called DeepDOF-SE that obtains microscopic images of tissue specimens, rapidly, with extended depth of focus to account for tissue surface irregularity, and virtual H&E staining to aid in interpretation. The technology is intended to be low cost so that it can potentially be utilized in low-resource settings. Development fresh tissue microscopes is an active area of research and the authors effectively combine/extend multiple previously demonstrated technologies to create a novel instrument that may be of some commercial interest due to its potential low cost. Results are impressive, the paper is nicely written, and the figures are easy to understand. DOF extension is simple, creative, results are very convincing and this advance will help a lot in real world utilization. The CycleGAN algorithm is very nice, should really advance the field by making it easier to train these types of networks. I am really excited about the potential cost-effectiveness of this technology, clear benefits in LMICs, but even in developed countries, this cost is getting close to the point where systems could be purchased without requiring capital equipment requests, meaning a much easier path to early

adoption. Minor critiques are listed below:

1. MUSE has some advantages (can be inexpensive) but also limitations. In particular, the fact that MUSE only images the very surface of the specimen (first few microns) brings adoption/utility challenges regarding distance of tumor from margin etc. Other issues like mucus and blood on the surface (even a very small amount) can confound the MUSE image. These potential issues and how they will be overcome or mitigated should be discussed.
2. Another important related issue with MUSE is the need for recuts, which are often recommended for primary diagnosis or to obtain additional/confirmatory information when a slide is equivocal.
3. There are other hardware microscope adaptations that can extend DOF, unclear why phase mask machine learning is superior or whether the combination can extend DOF further.
4. While it appears true that DeepDOF-SE provides superior results than conventional at defocus, the nuclear/intracellular detail at this NA is insufficient for both. While nuclei can be resolved, chromatin patterns cannot and that often is a critical component of nuclear histopathological diagnosis. It would be better if the NA were higher so that nuclear detail could be better visualized. This of course would affect the eventual DOF.
5. It would be good to see the OTFs of DeepDOF-SE and conventional. It looks like there is OTF loss in DeepDOF-SE especially when compared to conventional at 0 defocus.
6. Unclear as to whether MS-SSIM for DeepDOF-SE was referenced to the conventional or the DeepDOF-SE at 0 defocus.
7. The nuclear size of CycleGAN virtually stained slides seems to be significantly larger than H&E on the images but not backed up in supplementary data. Unclear as to why this is the case.
8. While the CycleGAN virtual staining process seems to work reasonably well, the staining pattern does diverge subtly from H&E. This divergence is probably not an issue for the examples shown but could be a major problem when trying to diagnose more subtle histologic features like an invasive squamous cell nest. There also seems to be a eosinophilic overflow - areas on the CycleGAN virtually stained slides that shouldn't be eosinophilic but are red and an understaining of nuclei - regions that should be basophilic are not as blue as they should be. Not clear if this is due to CycleGAN algorithm or MUSE or the stains/staining process. While partially addressed in supplementary information (difference in adipose tissue appearance), this issue still could be a major problem in practical utilization for more relevant and critical histopathological diagnostic features.
9. The text states that this tech "takes less than 10 minutes to stain and image a 7 cm² tissue sample." It would be helpful to get a more detailed breakdown on this timing.
10. Important to put in context that histopathology slides are generated in parallel - in other words a whole case is submitted at once but here each section needs to be done serially, so for larger cases where possible 20-40 cassettes are submitted, DeepDOF-SE processing will still take several hours. I still think DeepDOF-SE will win by a large margin, but worthwhile to think about the resources that will be required in even a small pathology department to convert to this new technology.

Response to Reviewers' comments:

Reviewer #1 (Remarks to the Author)

This is an important evolution for slide-free microscopy, and the combination of low cost and fast performance suggests that this could be widely adopted. My comments are not really sorted in terms of importance.

I recommend acceptance, after attention to some issues.

Comment 1: The images were acquired with a 4X, .13 NA objective. This makes capture of high-resolution details difficult--it would be helpful to hear what this implies for the proposed use case, as well as others.

Response 1: We specifically designed DeepDOF-SE with a 4X, 0.13 NA objective to provide a slide-free histology platform for use in low-resource settings to support immediate biopsy assessment and/or rapid intraoperative assessment of margin status. For this application, pathologists generally require only a 2X or 4X objective to provide definitive histopathologic diagnosis; thus, there is an important use case for the DeepDOF-SE system with a 4X objective. We revised the paper to make this use case clearer (see changes in lines 73 – 79, page 2) and included supporting references summarized below.

In the paper, we demonstrated that the subcellular features needed to diagnose pre-cancer and cancer¹, including architectural abnormalities, pleomorphism, abnormal nuclear morphology and increased nuclear-to-cytoplasmic ratio, are clearly resolved with a 4X objective (see **Fig. 7**, changes in lines 73-79 in paragraph 2 of the revised introduction, lines 416 – 421 in paragraph 3 of the revised discussion, and the new section in the supplemental “Objective Lens Design Choice”). We also demonstrated that pathologists could evaluate the features necessary to identify normal architecture and diagnose neoplasia (**new Fig. S11**). These changes are summarized below.

The proposed DeepDOF-SE use case is consistent with a recent study by Brown and colleagues, in which the pathologist’s use of 2X, 4X, 10X, and 20X objectives was monitored digitally during diagnostic review of H&E stained slides from a radical prostatectomy case². The pathologist used the 2X objective to examine 95% of the area. In comparison, the 10X objective was used to examine only 2% of the specimen area and the 20X objective was used to examine less than 0.15%. Similarly, in Mohs micrographic surgery, skin cancers are removed in stages until histologic examination of frozen sections confirms negative tumor margins³. Typically, frozen sections are examined with 1X-2X magnification^{4,5} which is adequate to assess margin status.

This use case is further supported by recent studies investigating the impact of image quality on diagnostic performance due to the increasing availability of high-resolution whole slide scanners⁶⁻⁹. In general, results show that accurate diagnosis can be obtained even without optimal image quality. For example, images can be significantly compressed (up to a compression ratio of 32:1) before impacting the ability of pathologists to interpret whole slide images⁶. A recent study examined the minimum resolution requirements of digital pathology images for accurate classification of breast cancer¹⁰. The investigators used large, publicly available datasets to compare diagnostic accuracy for deep learning classifier models as image resolution was computationally degraded. Results showed high diagnostic accuracy was maintained as NA was reduced from as high as 0.80-1.40 down to 0.05 (significantly lower than the 0.13 NA of DeepDOF-SE).

We agree that a higher NA (for example, at 10X, 0.30 NA or higher) is needed to resolve some sub-cellular features (e.g., vascular and perineural invasion, and for phenotypic characterization of tumor cells in undifferentiated malignancies)¹; however, this is typically not necessary for diagnosis of neoplasia.

Moreover, there are important tradeoffs associated with increasing the NA, including a smaller field of view and reduced depth of field. As previously shown by Baek et al., there exists a fundamental tradeoff between the transmission efficiency and depth invariance of an imaging system that has a modulated pupil function¹¹. We simulated the performance of DeepDOF-SE with a 10x 0.30 NA objective. New **Figure S3** (below) shows the Root Mean Square Error (RMSE) curve of simulated reconstruction (averaged over both fluorescence channels and 8000 images) across 21 defocus depths. Compared to the conventional baseline with the same objective lens, the DOF was significantly expanded from 7 microns to 40 microns (5.4x increase). However, this DOF is still far from the 200 microns required for imaging scalpel-cut irregular tissue surfaces. In this 40-micron DOF range, higher RMSE and decreased imaging performance were observed in defocus ranges of +/- 15 – 20 microns, making it challenging to resolve features in a target 200 um DOF range.

Thus, as we clarified in the revised paper, DeepDOF-SE with a 4X objective meets the performance requirements associated with the intended use case.

Comment 2: The CycleGAN was only trained on the same specimen in both DeepDOF-SE mode and H&E mode. This certainly encourage the model to perform well, but it would be instructive to see how it performs on other specimens of the same general type.

Response 2: We agree that it is critical to validate CycleGAN performance using external datasets. Following CycleGAN training with images of fresh human oral tumor, we evaluated model performance using images from three different tissue types and revised the paper accordingly as described below:

- 1) Images of frozen sections of mouse tongue. Results are shown in **Fig. 6** of the revised paper. While the CycleGAN was trained on fresh human oral tissue, we show both qualitatively in the figure and quantitatively in nuclear count that the cycleGAN performs well on mouse tongue tissue, which is of the same general tissue type but from a different species.
- 2) Images of other fresh human oral tumor resections not seen by the model during training. Results are shown in **Fig. 7** of the revised paper. Evaluated by an expert pathologist, the cycleGAN virtual staining of fresh oral tissue shows that the network performs well during inference.

3) Images of frozen sections of mouse esophagus. New **Figure S12** (right) shows an image of a mouse esophagus stained virtually using the CycleGAN algorithm. The image clearly shows the esophageal architecture with epithelium and surrounding connective tissue and muscle. As shown in selected ROIs, nuclei in the epithelial layer and connective tissue in the lamina propria in ROIs 1 and 2, as well as muscle fibers in ROIs 3 and 4, are visualized in the virtually stained DeepDOF-SE images.

Since CycleGAN virtual staining was trained using oral tissue images, we expect improved staining performance can be achieved in future work by training the model with an expanded image library of specific tissue types.

These three datasets support the ability of the model to accurately preserve nuclear/cytoplasmic features, and its capability to visualize features that are important for diagnosis of neoplasia.

Comment 3: There are some differences between DeepDOF-SE mode and H&E. One is illustrated in the main text but not discussed--I refer to the images shown in 7b in which the epidermis and dermis are quite different in appearance between DeepDOF-SE mode and H&E. Not fatal, but worth commenting on. Secondly, and more strikingly, the performance on fat (adipose) tissue with DeepDOF-SE mode is very different than in H&E--with fat showing up in bright red whereas it is clear (gone) in the H&E. This is an important performance difference that could have real implications. I strongly suggest that this example be moved to the main text and discussed. There should be technical solutions if this performance proves problematic.

Response 3: We agree that there are subtle differences between images acquired with DeepDOF-SE and H&E in **Figure 7b**. For example, collagen is more prominent and superficial keratin is stained more deeply in the conventional image of the H&E stained slide, and lymphocytes are more prominent in the DeepDOF-SE mode image. However, these differences are quite similar to variations in the intensity of staining that occur from lab to lab and daily within a single lab. Variations in factors such as the age of stains or the precise staining time can lead to intensity variations and overstaining issues in H&E stained slides¹². As a result, pathologists routinely review histopathology slides with staining variations. Despite the subtle differences between DeepDOF-SE images and H&E stained slides, epithelial architecture and cellular detail are clearly discerned in both, providing sufficient diagnostic information for clinical evaluation. We revised the paper to discuss these differences (see changes in lines 343 – 394 in paragraph one in section “H&E validation for DeepDOF-SE virtual staining in human oral tissue specimen”).

We agree with the reviewer that the appearance of adipose cells differs in DeepDOF-SE images compared to H&E stained slides. As we further clarified in the revised supplemental (see changes in lines 287 – 294 in paragraph one in section “Exceptional cases in CycleGAN results” in supplemental), this is primarily due to the staining of intact adipose cells in fresh tissue slices imaged with DeepDOF-SE. In contrast, the preparation of conventional H&E stained slides requires xylene exposure which removes lipids, giving

adipose cells a clear appearance. When it is necessary to stain lipid-containing structures, pathologists routinely use Oil Red O staining. Because DeepDOF-SE examines fresh tissue with intact lipids, the resulting lipid staining pattern is more similar to tissue stained with Oil Red O. Because this is a stain that is commonly used in pathology, we don't anticipate that it will result in interpretation challenges. In fact, it may be advantageous for evaluating certain tissue types; for example, in breast cancer, the presence of adipose cells is useful for delineating tumor margins.

Comment 4: Image 1B fluorescence panels could be brightened. Very hard to see properly.

Response 4: We increased the brightness of the fluorescence panels in revised **Figure 1b**.

Comment 5: What is the apparent depth of capture? In the dermis there seem to be about twice the number of cells that what shows up in the H&E equivalent. Worth a sentence or two.

Response 5: We thank the reviewer for raising an important point. In the revised manuscript, we clarified that the depth of capture using deep UV excitation is slightly deeper than a conventional section prepared using a microtome (4-10 microns). In a previous study¹³ on MUSE microscope by Fereidouni et al., it was reported that the number of nuclei present in fluorescence images using 280 nm excitation was slightly greater than that observed in images of conventional H&E stained sections, the extent to which depends on specimen tissue types. For example, in cranial nerve schwannoma, the nuclear count in deep-UV fluorescence images was 20% higher, while in cervical tissue it was about 60% higher than H&E stained sections. The authors noted that, although it may be necessary for pathologists to adjust the expected cellularity accordingly, the study demonstrated an accuracy of 94% - 100% for diagnosis of CNS lesions (n=24) by four pathologists using deep-UV fluorescence images compared to H&E stained slides. We revised the paper (lines 422 – 425 in paragraph 3 in discussion) to note the apparent increase in number of nuclei observed in DeepDOF-SE images, citing prior work.

Comment 6: The Supp. section seemed to imply that the DOF approach can work with RGB signals, not just 2-color, but it is not made clear what the story is or could be. Please explain more clearly.

Response 6: In **Figure S3**, we provided simulated results of the performance of DeepDOF-SE in RGB channels, even though the system was optimized for two fluorescence channels (473 nm and 640 nm). In this simulation, we explored the feasibility of using the optimized DeepDOF-SE design for broadband imaging beyond the two fluorescence channels. Our results show that the optimized design is highly achromatic, suggesting that the system is broadly compatible with other fluorescence dyes in the visible range. We describe this more fully in the first paragraph of the revised "Achromaticity of the DeepDOF-SE Design" section (lines 110 – 116 in Supplemental).

Comment 7: in focus: needs hyphen; Finetune: needs hyphen; ex vivo: needs hyphen; space needed between wavelength and nm, so 405 nm

Response 7: We made the requested changes throughout the revised manuscript.

Reviewer #2 (Remarks to the Author)

The work presented in the manuscript proposes a solution to the histopathological study of tissues obtained by resection during the development of surgery, avoiding the long times of histochemical preparation of standard procedures and the need for expensive infrastructures.

The solution proposed by the authors to achieve this purpose consists of a bi-channel fluorescence microscope with UV excitation that allows the observation of in situ samples obtained by scalpel resection and staining (DAPI and rhodamine B) in a rapid process of a few minutes. The image obtained is processed to obtain EDOF by a model based on a neural network trained from images with depth information, relying on a phase mask. This avoids the need to refocus the sample in a range of 200 microns. This range guarantees that all surface irregularities obtained by manual scalpel resection are correctly focused on the final image.

To facilitate the histopathological interpretation of the fluorescence images obtained, a virtual H&E staining method based on generative adversarial neural networks with cycle consistency (CycleGAN) is proposed. The proposed method consists of a semi-supervised training in two steps. In the first of them, the model is trained in a supervised way with a set of paired images obtained by applying analytical virtual staining (method based on the Beer-Lambert law) to the images of fluorescence. In the second step the model weights are fine-tuned through unsupervised training replacing the virtual staining images based on the Beer-Lambert law of the previous step with unpaired images of real standard H&E staining.

Therefore, the work presents a platform capable of performing slide-free histological studies suitable for rapid histopathological evaluation in intraoperative contexts or with limited resources based on bi channel fluorescence microscopy with superficial UV excitation. The platform is capable of scanning 1.6 cm²/min, with EDOF of 200 microns while preserving sub-cellular resolution.

The two most relevant aspects of the work presented refer to the processing of images obtained by UV excitation fluorescence microscopy (MUSE) are:

1. application of the EDOF method, proposed by the authors in the previous work Jin et al. (2020), to a bi channel fluorescence image obtained with DAPI and rhodamine B staining --in the previous work, the image was obtained by conventional single-channel fluorescence microscopy--.

2. H&E virtual staining of fluorescence images using a model based on generative adversarial neural networks with cycle consistency trained in two steps.

Regarding the two previous aspects, the work presents some important weaknesses. These can be summarized in that the manuscript should highlight the main contributions of the work presented and how these represent an advantage over the state of the art of the matter. More specifically:

Comment 1: Along with the proposed EDOF method, the authors should highlight what are the new contributions with respect to the previous published work (DOI: 10.1073/pnas.2013571117), beyond its application to a bi channel image. In addition, to judge the relevance of the results obtained it would be very enlightening to add a comparison of the alternative final performance with some other EDOF methods (e.g., z-stacking).

Response 1: In the revised manuscript, we further clarified the main contributions of the current work in the context of our previous publication and other EDOF such as z-stacking (see changes in lines 102 – 110 in paragraph 4 in the introduction; lines 404 – 409 in paragraph 3 in discussion):

- 1) Our previous work on DeepDOF focused on an end-to-end deep learning architecture to extend DOF in a single fluorescence channel. In contrast, our current work reports on a deep-learning-designed histology platform capable of generating H&E-like images of fresh tissue specimens at high speed and low cost. Compared to the previous work, our current work incorporates three key, novel components to make rapid and cost-effective histology practical: 1) surface excitation to enable optical sectioning, 2) deep-learning-enabled EDOF imaging in two fluorescence channels to simultaneously image nuclear and cytoplasmic histology, and 3) CycleGAN-based virtual staining to generate H&E-like histology images.
- 2) Other methods of extending DOF exist, primarily achieved via axial focus stacking. These methods leverage different mechanisms and instrumentation, such as a deformable mirror, a varifocal lens, or a digital micromirror device^{14–16}. Compared to these active axial sweeping approaches, DeepDOF-SE is less expensive (e.g., requiring only a \$10 phase mask) and allows for high throughput, single shot imaging. Reflectance confocal microscopy (RCM), for example, is used for skin cancer imaging with z-stacking; however, the system has a much lower scanning speed and is significantly more expensive (0.14 cm²/min and >\$100,000 for a Vivascope)^{17–20}. Full-field OCT is another modality that acquires z-stack information for histological diagnosis, and its scanning speed is about 1 cm²/min²¹.
- 3) The comparison of performance between z-stacking and the proposed DeepDOF-SE lies in the time it takes to scan the tissue, which is difficult to display in images. For the chosen 4x 0.13 NA objective, the original DoF is around 40 microns. Thus for each FoV, the z-stacking microscope will need to take at least 5 frames to cover the same depth-of-field as the proposed DeepDOF-SE. Not considering the time for motor movement, this is a 5x increase in capture time. The main advantage for z-stacking is that it does not depend on the optical properties of the objective and can extend the depth-of-field of the system to the full range of the light source. Since DeepDOF-SE is dependent on the objective's original depth-of-field, switching to a higher magnification objective will require z-stacking to achieve the full 200-micron range.

Comment 2: The virtual staining method should be accompanied by a discussion of related works that allow judging the relevance of contributions in this area. In addition, to justify the relevance of the proposed method, it is necessary its quantitative comparison with the result obtained with the virtual staining analytical method based on the Beer-Lambert law.

Response 2: We made changes in the revised paper (see lines 119 – 125, and 127 – 131 in paragraph 5 in introduction) to further clarify the relevance of contributions related to virtual staining. CycleGAN has been recently adapted for different imaging modalities, such as MUSE and photoacoustic microscopy, for the virtual staining of different tissue types, including brain, breast, prostate, and bone specimens. Compared to previous studies, the framework reported here is applicable to different staining protocols that provide both nuclear and cytoplasmic contrast. Furthermore, we report the first application of CycleGAN for virtual staining of fresh human oral tumor resections, and we demonstrate that our model is capable of visualizing distinct histological features in different layers of oral epithelium.

We also performed two additional experiments to quantitatively assess CycleGAN virtual staining compared to Beer-Lambert-law based method:

1) Comparison of two staining methods on images of frozen sections of mouse tongue in **Figure S6** (see updated figure in Supplemental). In Table S2, nuclear count and mean nuclear area were calculated using Cell Profiler; when comparing the mean nuclear area, CycleGAN virtual staining provides a closer match with standard H&E than Beer-Lambert-law based method, potentially due to the additional training of CycleGAN using standard H&E images.

2) A blinded review of images of freshly human oral cancer stained using two methods by two expert pathologists. In this pilot study, we demonstrate that the image quality scores of CycleGAN staining is higher than Beer-Lambert-law based method (more details are provided in the response to comment 3B from Reviewer 2).

Comment 3: Both the most important weaknesses mentioned previously, as well as the less important comments, have been contextualized as comments in the files (PDF) provided by the authors. [Authors note: Below are selected comments from PDF files]

Comment 3A: The reviewer commented on the chromatic aberration between the two fluorescence channels using the conventional microscope, asking if it is possible to provide an image or experiment that corroborates this assertion?

Response 3A: To demonstrate chromatic aberration between the two fluorescence channels, we imaged a frozen section of a mouse tongue stained with Rhodamine B and DAPI. New **Figure S4** (also shown here) shows images acquired using a conventional microscope at two axial planes that are 50 μm apart; the image in Rhodamine channel is in focus at axial plane 1, while the image in DAPI channel is in focus at axial plane 2. In contrast, as shown in **Figure 4**, DeepDOF-SE images in both channels are consistently in focus across the entire DOF range.

Comment 3B: I suggest providing a more detailed explanation of what the standard histopathological evaluation entails. To make this evaluation more meaningful (quantitative and/or qualitative), an evaluation experiment should be designed in which one or more pathologists assess different images without knowing their source (standard FFPE H&E, Beer-

Lambert-law-based method, and CycleGAN staining). This study should incorporate a statistical validation analysis.

Response 3B: We conducted a blinded review of histological features to further assess the diagnostic value of DeepDOF-SE images quantitatively. In this pilot evaluation, 20 DeepDOF-SE fluorescence images of fresh oral tumor resections, each containing varied histological features in a 2mm x 2mm FOV, were obtained. These fluorescence images were processed using the Beer-Lambert-law-based method and with CycleGAN staining, resulting in a total of 40 images each covering an FOV of 4 mm². De-identified images were presented in randomized order to two expert pathologists who were asked to

evaluate each image and assess the degree to which image quality was sufficient for 1) identification of architecture and normal structures, and 2) diagnosis of neoplasia. For each metric, three quality scores were used: 1=poor image quality, not sufficient for diagnosis, 2=moderate image quality, sufficient for diagnosis, and 3=good image quality, sufficient for diagnosis.

New **Figure S11** (also shown above) shows the mean image quality scores for the two image types; the mean image quality score was higher for images stained using CycleGAN for both pathologists. A Wilcoxon signed-rank test showed a significant difference ($Z = -2.55$, $p < 0.05$ for both metrics) between scores given for Beer-Lambert-law based images and CycleGAN images.

Comment 4: The assessment of the results obtained for EDOF deserves a more detailed discussion. Figure 4 shows the variation in MS-SSIM in a conventional fluorescence microscope and DeepDOF-SE when observing identical samples. In each case, the image obtained with the best possible focus (defocus: 0) is taken as the reference image. When comparing the images in the defocus: 0 column, it is visually apparent that the image from the conventional microscope is better focused than that of DeepDOF-SE. Could this fact lead to the comparisons in MS-SSIM loss not being properly normalized when comparing both microscopes?

Response 4: We used the image at 0 um defocus for the respective microscopes as the ground truth when comparing the MS-SSIM across different depths. The MS-SSIM between the conventional and DeepDOF-SE at 0 um defocus are as follows: Colon: 0.9039; esophagus: 0.8731; liver: 0.9001. As shown in new **Figure S6** (also shown here), the conventional image has a higher noise level than DeepDOF-SE, which contributes to the discrepancy in the MS-SSIM score at 0 um defocus. This is also consistent with our previous observation that the U-Net in DeepDOF-SE was trained with added noise and has been shown to have a denoising effect.

Comment 5: I suggest providing evidence to justify the need for using images outside the context of microscopy (natural scenes) to train this "multiple deconvolution" model.

Response 5: As described in the revised discussion section (lines 439 – 442 in paragraph 5 in discussion), the end-to-end extended DOF network aims to simulate physics-informed image formation and reconstruction, which are inherently insensitive to image content, and therefore, a data-agnostic approach was used for training. By incorporating natural scenes, we force the end-to-end DOF network to better sample the feature and frequency space, including sparse features that may be rarely present in tissue (e.g. straight lines and corners). This variability can help the model become more robust and adaptable to different types of inputs during inference, allowing it to generalize to a wider range of applications. Data fusion and dataset merging has also been used in other domains of imaging²²⁻²⁴.

Comment 6: It would be interesting to provide more information about the control of the X-Y scanning process and details regarding the scanning method and linear steps in X-Y.

Response 6: We provided more details regarding the scanning control process (see lines 478 – 481 in DeepDOF-SE Microscope Setup section).

Concluding Comment: After careful review of the manuscript and the supplementary material, I think that both require in-depth revision to address the issues that I have identified as particularly important. However, I do not want my opinions to overlook my sincere respect for the work done by the authors and the careful writing of the manuscript that facilitates its reading and understanding.

Reviewer #3 (Remarks to the Author)

In this manuscript entitled “DeepDOF-SE: affordable deep-learning microscopy platform for slide-free histology” by Jin et al., authors describe a technology called DeepDOF-SE that obtains microscopic images of tissue specimens, rapidly, with extended depth of focus to account for tissue surface irregularity, and virtual H&E staining to aid in interpretation. The technology is intended to be low cost so that it can potentially be utilized in low-resource settings. Development fresh tissue microscopes is an active area of research and the authors effectively combine/extend multiple previously demonstrated technologies to create a novel instrument that may be of some commercial interest due to its potential low cost. Results are impressive, the paper is nicely written, and the figures are easy to understand. DOF extension is simple, creative, results are very convincing and this advance will help a lot in real world utilization. The CycleGAN algorithm is very nice, should really advance the field by making it easier to train these types of networks. I am really excited about the potential cost-effectiveness of this technology, clear benefits in LMICs, but even in developed countries, this cost is getting close to the point where systems could be purchased without requiring capital equipment requests, meaning a much easier path to early adoption. Minor critiques are listed below:

Comment 1: MUSE has some advantages (can be inexpensive) but also limitations. In particular, the the fact that MUSE only images the very surface of the specimen (first few microns) brings adoption/utility challenges regarding distance of tumor from margin etc. Other issues like mucus and blood on the surface (even a very small amount) can confound the MUSE image. These potential issues and how they will be overcome or mitigated should be discussed.

Response 1: We revised the paper to clarify the following:

- 1) For intraoperative tumor margin assessment using DeepDOF-SE, resected surgical specimens will be first bread loafed with a scalpel into 3 – 4 mm thick transverse slices to access potential tumor margins on cross-sectional surfaces. This same process is routinely used to assess tumor margins using standard H&E slides prepared from frozen or permanent sections. In addition, we note that UV excitation with MUSE limits penetration depth to the first few microns, approximating the mechanical sectioning performed with a microtome in standard preparation of H&E stained slides.
- 2) As described in the revised “Tissue Processing and Imaging” section (lines 594 and 607), fresh tissue slices (3-4 mm thick) are gently rinsed with PBS prior to vital dye staining to remove mucus and blood on the surface.

Comment 2: Another important related issue with MUSE is the need for recuts, which are often recommended for primary diagnosis or to obtain additional/confirmatory information when a slide is equivocal.

Response 2: We agree that recuts are sometimes necessary to obtain additional information when a slide is equivocal. Because DeepDOF-SE provides images rapidly, fresh samples can be immediately manipulated to provide additional information in three ways: 1) the opposite side of the slice can be imaged; 2) the slice can be recut with a scalpel in 2-3 mm steps; or 3) the sample could be submitted for frozen or permanent section in equivocal cases. We revised the paper to discuss this limitation and possible solutions (see lines 396 – 400 in paragraph 2 in discussion).

Comment 3: There are other hardware microscope adaptations that can extend DOF, unclear why phase mask machine learning is superior or whether the combination can extend DOF further.

Response 3: We agree with the reviewer that other hardware microscope adaptations exist to extend DOF. Most previous efforts to extend DOF have used axial focus stacking, leveraging different mechanisms and instrumentation, such as a deformable mirror, a varifocal lens, and a digital micromirror device^{14–16}. Compared to these active axial sweeping approaches, our reported method of a deep-learning-enabled phase mask has several advantages. First, the phase mask in DeepDOF-SE is passive and can acquire an all-in-focus image in a single shot, thus allowing rapid image acquisition. Second, the phase mask is inexpensive (<\$10), passive, and readily compatible with existing fluorescence microscope. These advantages are summarized in revised paragraph 3 of the discussion (lines 404 – 409). Finally, we demonstrated that our deep-learning-enabled phase mask design outperforms conventional wavefront encoding approaches such as a cubic phase mask.

We previously demonstrated that the combination of phase mask and deep learning can extend the DOF further. In our evaluation in **Figure R1**, we adapted the end-to-end DOF network to optimize performance to extend DOF to 300 and 400 μm at 550 nm, showing significant improvement over a conventional microscope at 0.13 NA and 0.06 NA. However, we also note there exists a tradeoff between the overall RMSE performance and the DOF range.

Comment 4: While it appears true that DeepDOF-SE provides superior results than conventional at defocus, the nuclear/intracellular detail at this NA is insufficient for both. While nuclei can be resolved, chromatin patterns cannot and that often is a critical component of nuclear histopathological diagnosis. It would be better if the NA were higher so that nuclear detail could be better visualized. This of course would affect the eventual DOF.

Response 4: We designed DeepDOF-SE with a 4X objective to provide a slide-free histology platform for use in low-resource settings to support immediate biopsy assessment and/or rapid intraoperative assessment of margin status. For this application (as described in greater detail in response to Comment 1 from Reviewer 1), pathologists generally require only a 2X or 4X objective to provide definitive histopathologic diagnosis^{2,4,5}; thus, there is an important use case for the DeepDOF-SE system with a 4X objective. We revised the paper to make this use case clearer; see Response 1 to Reviewer 1 for a more detailed description of these changes. Briefly, we showed that the subcellular features needed to diagnose pre-cancer and cancer¹, including architectural abnormalities, pleomorphism, and abnormal nuclear morphology and increased nuclear-to-cytoplasmic ratio (**Fig. 7**), are clearly resolved with a 4X objective (see changes in lines 416 – 421 in paragraph 3 in discussion). We agree that a higher NA (for example, at 10X, 0.30 NA or higher) is needed to resolve some sub-cellular features (e.g., vascular and perineural invasion, and for phenotypic characterization of tumor cells in undifferentiated malignancies)¹; however, this is typically not necessary for diagnosis of pre-cancer or cancer.

Figure R1. The end-to-end DOF framework can be used to further improve the DOF beyond 200 μm using a NA 0.13 objective.

Moreover, there are important tradeoffs associated with increasing the NA, including a smaller field of view and reduced depth of field. As previously shown by Baek et al.¹¹, there exists a fundamental tradeoff between the transmission efficiency and depth invariance of an imaging system that has a modulated pupil function. We simulated the performance of DeepDOF-SE with a 10x 0.30 NA objective. New **Figure S3** shows the Root Mean Square Error (RMSE) curve of simulated reconstruction (averaged over both fluorescence channels and 8000 images) across 21 defocus depths. Compared to the conventional baseline with the same objective lens, the DOF was significantly expanded from 7 microns to 40 microns (5.4x increase). However, this DOF is still far from the 200 microns required for imaging scalpel-cut irregular tissue surfaces. In this 40-micron DOF range, higher RMSE and decreased imaging performance were observed in defocus ranges of +/- 15 – 20 microns, making it challenging to resolve features in a target 200 um DOF range.

In addition to the theoretical limit of how much one can extend the depth-of-field without introducing mechanical z-scanning into the system, we also considered the field-of-view of the objectives when selecting the 4X objective. The table below compares various features of 4X, 10X, and 20X objectives and how they would impact the scanning speed. Since DeepDOF-SE is designed to provide rapid assessment of histology at the point of care, with the potential for permanent FFPE slides to be made when needed, we prioritized fast scanning speed with good resolution and contrast over providing the highest magnification.

Calculated for 550 nm	4x, 0.13 NA	10x, 0.3 NA	20x, 0.5 NA
Resolution	2.58 um	1.12 um	0.67 um
FoV (Diameter)	6.62 mm	2.65 mm	1.3 mm
DoF	40.2 um	7.4 um	2.6 um
# of frames for 1 cm x 1 cm x 200 um with z- stacking and no lateral overlap	45	972	9,317

Comment 5: It would be good to see the OTFs of DeepDOF-SE and conventional. It looks like there is OTF loss in DeepDOF-SE especially when compared to conventional at 0 defocus.

Response 5: Figure S5 (also shown below) shows the updated modulated transfer function figure for both DeepDOF-SE and a conventional microscope. The close proximity of MTFs in 3 color channels shows that the end-to-end optimized system is highly achromatic. All 3 color channels achieved high contrast in the majority of frequency ranges within the DOF. In contrast, the conventional microscope's MTF shows rapid decrease in area under the MTF curve as the defocus increases. The effects of chromatic aberration can also be observed in the separation of MTF curves. However, we note that the forward optical model does not fully capture the aberration caused by the objective and tube lens of the experimental system.

There is MTF loss in the DeepDOF-SE since the USAF target we used to generate the MTF contains line pairs beyond the diffraction limit of the objective. **Figure R2** shows a comparison between the conventional microscope and the DeepDOF-SE reconstruction of the USAF target used. Only the blue channel is shown here. The DeepDOF-SE reconstruction shows better contrast and less noise than the conventional capture. The contrast of DeepDOF-SE here is better than the experimental USAF shown in **Figure 3** since the PSF does not suffer from any fabrication or calibration errors.

Comment 6: Unclear as to whether MS-SSIM for DeepDOF-SE was referenced to the conventional or the DeepDOF-SE at 0 defocus.

Response 6: MS-SSIM was referenced to their respective 0 micron defocus. The MS-SSIM between the conventional and DeepDOF-SE are 0.9039 (colon), 0.8731 (esophagus), 0.9001 (liver). We also added descriptions in greater detail in response to Comment 4 from Reviewer 2.

Comment 7: The nuclear size of CycleGAN virtually stained slides seems to be significantly larger than H&E on the images but not backed up in supplementary data. Unclear as to why this is the case.

Response 7: In the original manuscript, we compared images obtained with DeepDOF-SE using a 4X objective and images of standard H&E slides obtained with a 10X objective. In the revised manuscript, we down-sampled images of standard H&E slides to approximate the magnification of the DeepDOF-SE, as shown in the updated **Figures 6** and **Figure S9**. In addition, we refined the segmentation maps in **Figure S9** using Cell Profiler, and we also updated the nuclear count and nuclear area metrics in **Table S2** accordingly.

Comment 8: While the CycleGAN virtual staining process seems to work reasonably well, the staining pattern does diverge subtly from H&E. This divergence is probably not an issue for the examples shown but could be a major problem when trying to diagnose more subtle histologic features like an invasive squamous cell nest. There also seems to be a eosinophilic overflow - areas on the CycleGAN virtually stained slides that shouldn't be eosinophilic but are red and an understaining of nuclei - regions that should be basophilic are not as blue as they should be. Not clear if this is due to CycleGAN algorithm or MUSE or the stains/staining process. While partially addressed in supplementary information (difference in adipose tissue appearance), this issue still could be a major problem in practical utilization for more relevant and critical histopathological diagnostic features.

Response 8: We agree with the reviewer that there are subtle differences in the appearance of DeepDOF-SE images and conventional images of H&E stained slides. As previously discussed in our response to Reviewer 1 Comment 3, we revised the paper to better describe the well-known variations in the intensity of H&E staining between different pathology labs and from day-to-day within individual pathology labs, often due to differences in the age of stains and reagents or the precise amount of time the slide is exposed to each reagent. This typically results in intensity variations and sometimes overstaining or understaining issues in H&E. In **Fig. 6c**, for example, we observed that basophilic regions in H&E stained frozen sections are not as blue as the permanent sections; in comparison, these regions appear more blue in DeepDOF-SE images.

The eosinic overflow in the cycleGAN results in **Figure 6** might be a result of excess Rhodamine B on the glass slide. The Beer-Lambert staining of the input image also has red specks which are not present on the permanent H&E. The frozen section slides cannot be rinsed as thoroughly as the fresh tissue sections (sectioned tissue can lift off the slide during rinsing with PBS). Additionally, we improved the process of dissolving Rhodamine B in the solvent to reduce this residual problem.

Comment 9: The text states that this tech “takes less than 10 minutes to stain and image a 7 cm² tissue sample.” It would be helpful to get a more detailed breakdown on this timing.

Response 9: We further clarified the time breakdown in the revised paper (line 388 in paragraph 2 in discussion). Specifically, to image a 7 cm² tissue, the staining time is 4 min and the scanning time is <5 min at a scanning speed of 1.6 cm²/min.

Comment 10: Important to put in context that histopathology slides are generated in parallel - in other words a whole case is submitted at once but here each section needs to be done serially, so for larger cases where possible 20-40 cassettes are submitted, DeepDOF-SE processing will still take several hours. I still think DeepDOF-SE will win by a large margin, but worthwhile to think about the resources that will be required in even a small pathology department to convert to this new technology.

Response 10: We agree it is important to compare the overall workflow of traditional histopathology and DeepDOF-SE in the context of an entire case. The table below compares the workflow of traditional histopathology and DeepDOF-SE. For both approaches, the workflow includes two parts: 1) Sample preparation and 2) Image acquisition and review. DeepDOF-SE can significantly reduce both time and infrastructure needed to process an entire case.

	Traditional Histopathology	DeepDOF-SE
1. Sample preparation methods	Breadloafing: Fresh tissue examined by a pathologist, breadloafed, slices are selected for frozen section, and remainder of case is submitted for FFPE. Frozen section: Histotechnologist embeds selected slices in OCT, freezes the sample, and prepares 10-20 um thick slices using a cryostat microtome. Slides are stained with H&E using an automated slide stainer. FFPE: Histotechnologist places slices in cassettes that are embedded in paraffin. A microtome is used to obtain 4-6 um thick slices. Slides are stained with H&E using an automated slide stainer.	Breadloafing: Fresh tissue is, breadloafed. Staining: Histotechnologist immerses all slices in solution containing stain, slices are rinsed, and placed on the microscope stage
Infrastructure for sample preparation	Breadloafing: Sharp knife Frozen section: Cryostat microtome & supplies FFPE: Microtome & supplies H&E: Automated slide stainer & reagents	Breadloafing: Sharp knife Staining: Staining solution, saline rinse bottle
Sample preparation time	Total processing time for a single frozen section is ~20 min. Busy clinics often have multiple histotechnologists and cryostat microtomes to process slices in parallel but this increases infrastructure costs. Additional time is required to prepare FFPE slides for later review.	Total processing time for a single slice is <1 min. All slices can be processed in parallel. If desired FFPE slides can be prepared for later review.
2. Image acquisition and review	The total time for image acquisition and review is the same for traditional histopathology and DeepDOF-SE.	

Infrastructure for image acquisition and review	Conventional white light microscope. Digital images can be analyzed using automated algorithms	Fluorescence microscope with \$10 phase mask. Digital images can be analyzed using automated algorithms
---	---	--

References

1. Robbins, S. L. (Stanley L. *Robbins & Cotran pathologic basis of disease*. (Elsevier, 2021).
2. Ashman, K. *et al.* A Camera-Assisted Pathology Microscope to Capture the Lost Data in Clinical Glass Slide Diagnosis. 2022.08.31.506042 Preprint at <https://doi.org/10.1101/2022.08.31.506042> (2022).
3. Wong, E., Axibal, E. & Brown, M. Mohs Micrographic Surgery. *Facial Plast. Surg. Clin. N. Am.* **27**, 15–34 (2019).
4. Sutton, E. & Hanke, C. W. Microscope Use in Mohs Micrographic Surgery: A Survey of Current and Former Mohs Surgery Fellowship Directors. *Dermatol. Surg. Off. Publ. Am. Soc. Dermatol. Surg. Al* **48**, 786–787 (2022).
5. Patel, Y. G. *et al.* Confocal reflectance mosaicing of basal cell carcinomas in Mohs surgical skin excisions. *J. Biomed. Opt.* **12**, 034027 (2007).
6. Krupinski, E. A., Johnson, J. P., Jaw, S., Graham, A. R. & Weinstein, R. S. Compressing pathology whole-slide images using a human and model observer evaluation. *J. Pathol. Inform.* **3**, 17 (2012).
7. Mukhopadhyay, S. *et al.* Whole Slide Imaging Versus Microscopy for Primary Diagnosis in Surgical Pathology: A Multicenter Blinded Randomized Noninferiority Study of 1992 Cases (Pivotal Study). *Am. J. Surg. Pathol.* **42**, 39–52 (2018).
8. Goacher, E., Randell, R., Williams, B. & Treanor, D. The Diagnostic Concordance of Whole Slide Imaging and Light Microscopy: A Systematic Review. *Arch. Pathol. Lab. Med.* **141**, 151–161 (2017).
9. Kim, H. *et al.* Deep learning-based histopathological segmentation for whole slide images of colorectal cancer in a compressed domain. *Sci. Rep.* **11**, 22520 (2021).
10. Neary-Zajiczek, L. *et al.* Minimum resolution requirements of digital pathology images for accurate classification. *Med. Image Anal.* **89**, 102891 (2023).
11. Baek, J. Transfer efficiency and depth invariance in computational cameras. in *2010 IEEE International Conference on Computational Photography (ICCP)* 1–8 (IEEE, 2010). doi:10.1109/ICCPHOT.2010.5585098.
12. An Intro to H&E Staining: Protocol, Best Practices, Steps & More. <https://www.leicabiosystems.com/us/knowledge-pathway/he-staining-overview-a-guide-to-best-practices/>.
13. Fereidouni, F. *et al.* Microscopy with ultraviolet surface excitation for rapid slide-free histology. *Nat. Biomed. Eng.* **1**, 957–966 (2017).
14. Chen, Y., Liu, H., Zhou, Y., Kuang, F.-L. & Li, L. Extended the depth of field and zoom microscope with varifocal lens. *Sci. Rep.* **12**, 11015 (2022).
15. Xiao, S., Tseng, H., Gritton, H., Han, X. & Mertz, J. Video-rate volumetric neuronal imaging using 3D targeted illumination. *Sci. Rep.* **8**, 7921 (2018).

16. Shain, W. J., Vickers, N. A., Goldberg, B. B., Bifano, T. & Mertz, J. Extended depth-of-field microscopy with a high-speed deformable mirror. *Opt. Lett.* **42**, 995–998 (2017).
17. Kose, K. *et al.* Video-mosaicing of reflectance confocal images for examination of extended areas of skin in vivo. *Br. J. Dermatol.* **171**, 1239–1241 (2014).
18. Edwards, S. J. *et al.* VivaScope® 1500 and 3000 systems for detecting and monitoring skin lesions: a systematic review and economic evaluation. *Health Technol. Assess. Winch. Engl.* **20**, 1–260 (2016).
19. Rajadhyaksha, M., Marghoob, A., Rossi, A., Halpern, A. C. & Nehal, K. S. Reflectance confocal microscopy of skin in vivo: From bench to bedside. *Lasers Surg. Med.* **49**, 7–19 (2017).
20. TKACZYK, E. R. Innovations and Developments in Dermatologic Non-invasive Optical Imaging and Potential Clinical Applications. *Acta Derm. Venereol. Suppl* **218**, 5–13 (2017).
21. Thouvenin, O., Grieve, K., Xiao, P., Apelian, C. & Boccara, A. C. En face coherence microscopy [Invited]. *Biomed. Opt. Express* **8**, 622–639 (2017).
22. Meng, T., Jing, X., Yan, Z. & Pedrycz, W. A survey on machine learning for data fusion. *Inf. Fusion* **57**, 115–129 (2020).
23. Lambert, J., Liu, Z., Sener, O., Hays, J. & Koltun, V. MSeg: A Composite Dataset for Multi-domain Semantic Segmentation. Preprint at <https://doi.org/10.48550/arXiv.2112.13762> (2021).
24. Zhang, W. *et al.* Merging nucleus datasets by correlation-based cross-training. *Med. Image Anal.* **84**, 102705 (2023).

Reviewers' Comments:

Reviewer #1:

Remarks to the Author:

The responses to the reviewers and the changes to the submitted manuscript are appropriate. No further alterations are required, and I recommend acceptance.

Reviewer #2:

Remarks to the Author:

Having carefully reviewed the authors' responses to the various reviewers' comments and the revised version of the manuscript. I am pleased to convey my positive assessment of the work.

The authors have demonstrated a commendable commitment to addressing all aspects raised by the reviewers. Their responses are articulate, comprehensive, and provide clarity on the various points discussed. The revised manuscript now incorporates all requested clarifications and precision, displaying the authors' responsiveness to feedback.

I would like to commend the authors for their ability to express their ideas in a clear and rigorous manner. The revised manuscript reflects not only a commitment to addressing concerns but also a dedication to maintaining a high standard of clarity and academic rigour.

Considering the thorough and effective revisions made, I have no hesitation in recommending the publication of the revised version of the manuscript.

Reviewer #3:

Remarks to the Author:

The revised manuscript is significantly improved and the authors have satisfactorily addressed all of my comments and questions.